# Evolving Interdependent Operators with Large Language Models for Multi-Objective Combinatorial Optimization

**Junhao Qiu** [1]  **Xin Chen** [1]  **Liang Ge** [2]  **Liyong Lin** [2]  **Zhichao Lu** [1]  **Qingfu Zhang** [1]

## Abstract

Neighborhood search operators are critical to the performance of Multi-Objective Evolutionary Algorithms (MOEAs) and rely heavily on expert design. Although recent LLM-based Automated Heuristic Design (AHD) methods have made notable progress, they primarily optimize individual heuristics or components independently, lacking explicit exploration and exploitation of dynamic coupling relationships between operators. In this paper, multi-operator optimization in MOEAs is formulated as a Markov decision process, enabling the improvement of interdependent operators through sequential decision-making. To address this, we propose the **E**volution **o**f **O**perator **C**ombination (**E2OC**) framework for MOEAs, which achieves the co-evolution of *design strategies* and executable *codes*. E2OC employs Monte Carlo Tree Search to progressively search combinations of operator design strategies and adopts an operator rotation mechanism to identify effective operator configurations while supporting the integration of mainstream AHD methods as the underlying designer. Experimental results across AHD tasks with varying objectives and problem scales show that E2OC consistently outperforms state-of-the-art AHD and other multi-heuristic co-design frameworks, demonstrating strong generalization and sustained optimization capability.

## 1. Introduction

Multiobjective Combinatorial Optimization Problems (MCOPs) are widely encountered in fields such as production scheduling (Neufeld et al., 2023; Li et al., 2024b),

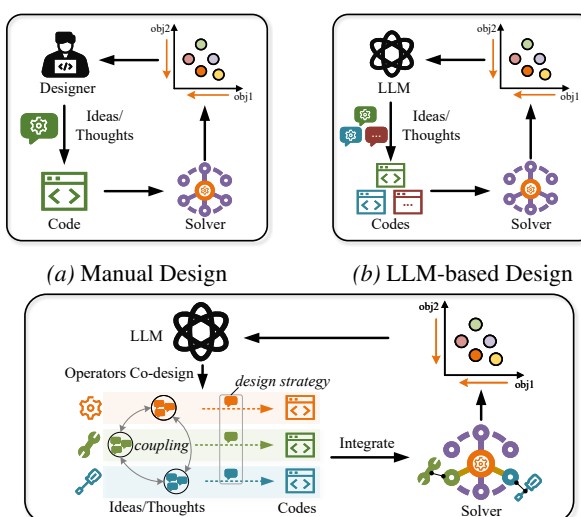

*(a)* Manual Design    *(b)* LLM-based Design

*(c)* LLM-based Multi-Operator Co-design

*Figure 1.* The single operator design in MOEAs has advanced from (a) expert-dependent methods to (b) LLM-guided iterative improvements of code implementations (e.g., EoH, MCTS-AHD). In comparison, (c) E2OC explicitly accounts for interdependencies among mult-operators and facilitates the coordinated co-evolution of *design strategies* and executable *codes*.

engineering design (Peng et al., 2023), and hyperparameter tuning in machine learning (Morales-Hernández et al., 2023). For these NP-hard problems, obtaining the entire Pareto set/frontier using exact algorithms (e.g., dynamic programming) is challenging (Wang et al., 2023). Meta heuristic-based approximation approaches include Multi-Objective Evolutionary Algorithms Multi-Objective Evolutionary Algorithms (MOEAs) (e.g., NSGA-II (Deb et al., 2002), NSGA-III (Deb & Jain, 2014), MOEA/D (Qingfu Zhang & Hui Li, 2007)), Pareto local search methods (e.g., PLS (Paquete et al., 2004b), 2PPLS (Lust & Teghem, 2010a), PPLS/D-C (Shi et al., 2022)), and methods combining evolutionary algorithms and local search (Paquete et al., 2004a; Kumar & Singh, 2007; Jaszkiewicz & Zielniewicz, 2009). However, the effectiveness of MOEAs depends on the selection and interaction of domain-specific search operators. Different application domains typically require different algorithms and/or configurations. Manually designing

---

[1]Department of Computer Science, City University of Hong Kong [2]Contemporary Amperex Technology Limited. Correspondence to: Liyong Lin <llin5@e.ntu.edu.sg>, Qingfu Zhang <qingfu.zhang@cityu.edu.hk>.

*Proceedings of the 43rd International Conference on Machine Learning*, Seoul, South Korea. PMLR 306, 2026. Copyright 2026 by the author(s).

and tuning these operators is costly and heavily reliant on expert knowledge.

Automated Heuristic Design (AHD) is a promising research direction for addressing this problem (Burke et al., 2013; Stützle & López-Ibáñez, 2019). Genetic Programming (GP), one of the earliest techniques for automatic heuristic discovery (**?**Zhang et al., 2023), evolves algorithms via simulated natural selection. However, GP relies on a set of permissible primitives or mutation operations; constructing a domain-agnostic set remains fundamentally difficult across diverse multiobjective metaheuristics and problem settings (Pillay & Qu, 2018; O'Neill et al., 2010).

The AHD powered by the code generation and language comprehension capability of the Large Language Models (LLMs) has introduced a new search paradigm in recent years (Liu et al., 2026; Wu et al., 2024). LLMs are employed as a heuristic designer in certain iterative frameworks (Zhang et al., 2024b), such as evolutionary search (Liu et al., 2024a; Ye et al., 2024; van Stein & Bäck, 2024; Yao et al., 2025), neighborhood search (Xie et al., 2025), and Monte Carlo Tree Search (MCTS) (Zheng et al., 2025b; Kiet et al., 2025). These methods represent a shift from traditional approaches, leveraging LLMs' reasoning abilities to synthesize algorithmic ideas and adapt them to problem-specific tasks. While these methods have achieved significant progress in evolving single operator, they focus primarily on evolving isolated components rather than multi-operator systems.

Effective optimization requires combining operators with complementary search biases in complex MCOPs. The overall performance of MOEAs therefore depends on how well these operators complement and interact with one another to balance exploration and exploitation throughout the optimization process. However, both expert-designed and LLM-driven approaches, particularly traditional single-operator evolution methods, remain limited in systematically reasoning about operator interactions and sequencing effects. Consequently, establishing a co-evolutionary mechanism for operators is essential to enhancing the performance and adaptability of MOEAs in multi-objective optimization.

To bridge this gap, we propose a new algorithm design paradigm, dubbed the **E**volution **of** **O**perator **C**ombination (**E2OC**). This approach searches for *design strategy* formed by multiple design thoughts of different operators, modeling their interdependencies and synergies to guide the evolution of operator combinations. The design thoughts are textual descriptions of specific improvement suggestions intended to enhance the existing operators, rather than merely expressing the semantic idea of an operator (Liu et al., 2024a). By systematically exploring combinations of these thoughts, the search for effective operator combinations is guided in promising directions. We demonstrate that LLM-assisted co-evolution of design thoughts and executable codes, guided by carefully crafted prompts, achieves state-of-the-art multi-operator design results. We expect E2OC to provide a significant advancement in the automated design of complex algorithms. In summary, our contributions are as follows:

- We propose E2OC, a new algorithm design paradigm that supports the LLM-based co-evolution of *design strategies* and *codes*, achieving automated design of multi-operators in MOEAs with minimum hand-craft.

- We develop a progressive design strategy search mechanism that explores the coupling between operators by combining the design thoughts of different operators to guide the direction of evolution.

- We implement operator rotation evolution to systematically explore design strategies and identify optimal operator combinations, while supporting the integration of advanced AHD methods as algorithm designer.

- We comprehensively evaluate E2OC on benchmarks of two widely studied MCOPs. The results demonstrate that E2OC outperforms many existing AHD methods, such as EoH and MCTS-AHD. In two- or three-objective problems, E2OC brings significant enhancements in manually designed MOEAs. In particular, E2OC is able to leverage the evolution of design strategies to achieve sustained enhancements under continued resource investment.

## 2. Multi-Operator Optimization in MOEAs

Based on previous designs of single heuristics (Liu et al., 2024a; Romera-Paredes et al., 2024), recent work has been extended to more complex algorithmic systems, such as heuristic set design (Liu et al., 2025) and multi-strategy optimization (Kiet et al., 2025). In contrast, we focus on the co-design of interdependent operators in MOEAs that involve evaluation uncertainty.

**Multi-Objective Optimization.** The task of multi-objective optimization is to identify a set of solutions that balance multiple, often conflicting, objectives. A general formulation is given by:

$$\min_{\boldsymbol{x} \in \mathcal{X}} \boldsymbol{f}(\boldsymbol{x}) = (f_1(\boldsymbol{x}), f_2(\boldsymbol{x}), \ldots, f_M(\boldsymbol{x})), \qquad (1)$$

where $\mathcal{X}$ denotes the feasible solution space, $\boldsymbol{x}$ is a decision vector, and $\boldsymbol{f} : \mathcal{X} \to \mathbb{R}^M$ represents an objective function vector with $M$ objectives.

**Pareto Dominance.** Given two solutions $\boldsymbol{x}_a, \boldsymbol{x}_b \in \mathcal{X}$, $\boldsymbol{x}_a$ is said to dominate $\boldsymbol{x}_b$ (denoted $\boldsymbol{x}_a \prec \boldsymbol{x}_b$) if and only if $f_i(\boldsymbol{x}_a) \leq f_i(\boldsymbol{x}_b), \forall i \in \{1, 2, \ldots, M\}$, and $f_j(\boldsymbol{x}_a) < f_j(\boldsymbol{x}_b)$ for at least one $j$.

**Pareto Optimality:** A solution $x^* \in \mathcal{X}$ is Pareto-optimal if there is no $x' \in \mathcal{X}$ such that $x' \prec x^*$. In other words, no feasible solution exists that can improve one objective without degrading another. The set of all Pareto-optimal solutions in the decision space is defined as Pareto set, whose mapping in the objective space yields the Pareto front (PF).

**Multi-Operator Solver.** The performance of MOEAs depends on the operators or search strategies employed, including their actions and parameter settings. We define a solver $\mathcal{S}(d \mid \boldsymbol{O})$ parameterized by a combination of $K$ operators $\boldsymbol{O} = (O_1, O_2, \ldots, O_K)$, which generates candidate solution set $\mathcal{A} \subset \mathcal{X}$ for a specific problem instance $d$:

$$f(\mathcal{A}) = \{f(x) \mid x \in \mathcal{A}\}. \tag{2}$$

Each operator $O_i$, $i \in 1, \ldots, K$ is instantiated through the algorithm generator $\mathcal{G}(\cdot \mid p_i)$ guided by a prompt $p_i$ containing the reference information about the code template of $O_i$. Prompts are generated by the prompt generator $\mathcal{P}(\cdot \mid O_i)$, and all prompts form the tuple $\boldsymbol{p} = (p_1, p_2, \ldots, p_K)$. Operators act on specific decision subspaces and perform search operations to optimize solutions.

**Operator Combination Optimization.** Each operator $O_i$ in the operator combination can be iteratively refined through evolutionary updates to enhance the solver's overall performance. To obtain a scalar performance value from a multi-objective evaluation, the performance of the solution set is evaluated by an aggregation function $\Phi(\cdot)$, mapping it to a scalar in $\mathbb{R}$:

$$F(d \mid \boldsymbol{O}) = \Phi(f(\mathcal{S}(d \mid \boldsymbol{O}))) \tag{3}$$

This function maps the multi-objective performance set into a single-valued score, where $\Phi(\cdot)$ can represent metrics such as the Hypervolume (HV) (Zitzler & Thiele, 1999), the Inverted Generational Distance (IGD) (Coello & Cortés, 2005)). To reduce stochastic variance, multiple independent evaluations are conducted. Given $N$ evaluations on instance $d$, the performance of the $n$-th evaluation is doneted as $F^n(d \mid \boldsymbol{O})$. The averaged performance with $N$ evaluations under the solver $\mathcal{S}$ parameterized by $\boldsymbol{O}$ is calculated as:

$$\bar{F}^N(d \mid \boldsymbol{O}) = \frac{1}{N} \sum_{n=1}^{N} F^n(d \mid \boldsymbol{O}) \tag{4}$$

The operator combination $\boldsymbol{O}$ belongs to the multi-operator space $\mathbf{S}$, and $\boldsymbol{O}$ can be sampled using algorithm and prompt generators. Assuming instance set $\mathcal{D}$, the overall optimization objective given a limited budget is formulated as:

$$\boldsymbol{O}^* \in \arg\max_{\boldsymbol{O} \in \mathbf{S}} \mathbb{E}_{d \sim \mathcal{D}} \left[ \bar{F}^N(d \mid \boldsymbol{O}) \right], \tag{5}$$

Although this formulation models operator combination evolution as a higher-level optimization and accounts for their collective contribution, it does not explicitly capture the interdependencies among operators during the search process. These dynamic dependencies can be further analyzed through a Markov Decision Process (see **Appendix C**).

## 3. Methodology

### 3.1. Overall Framework of E2OC

E2OC designs multi-operators in MOEAs automatically by using LLM to co-evolve operator combinations and design strategies. The overall framework is shown in Figure 2, including four core components: **1) Warm-start:** the algorithm generator $\mathcal{G}(\cdot \mid \boldsymbol{p}_0)$ is employed to generate candidate operator combination set $OS$ with a prompt containing the initial prompt template tuple $\boldsymbol{p}_0$ included code templates. Then the elite operators are then analyzed and summarized using the prompt generator $\mathcal{P}(\cdot)$ to extract different design thoughts. **2) Language space of design ideas:** The multi-domain design thoughts extracted from elite operators constitute the language space of operator design strategies, with complex cross-domain coupling relationships. **3) Progressive search for design strategy:** Different combinations of design thoughts in the language space are explored and evaluated by MCTS to locate the best potential strategy. **4) Multi-Operator Design and Evaluation:** The operators in the operator combinations are designed sequentially rotating one by one based on the design strategy. The newly generated operator combinations will be integrated in MOEAs to evaluate the performance and the scores will be used to update the branching information of the Monte Carlo tree.

E2CO combines different interdependent operator design thoughts to achieve a co-evolution of the design strategies and executable codes. Notably the specific operator improvement suggestions in the design strategy are integrated into the prompts for evolution. Unlike modifying prompts directly with LLM, it enables the exploration and optimization of design knowledge at semantic level.

### 3.2. Warm-Start Initialization of Multi-Operator Sets

Different operators in MOEAs have their own independent coding domains and neighborhood structures, and their effective co-designs are often characterized as strong coupling, diverse and weakly separable. Primarily, E2OC performs independent evolution and knowledge extraction for each operator $O_i$, $i \in 1, \ldots, K$ (see Algorithm 1):

**Step 1:** For different operator $i$, the candidate operators are generated by the algorithm generator $\mathcal{G}_i(\cdot \mid p_i)$ based on initial prompts $p_i \in \boldsymbol{p}_0$. And the operators are checked for validity to remove illegal and invalid operators.

**Step 2:** Initialize the multi-operator solver $\mathcal{S}(\cdot)$ with the optimization problem to construct the multi-objective eval-

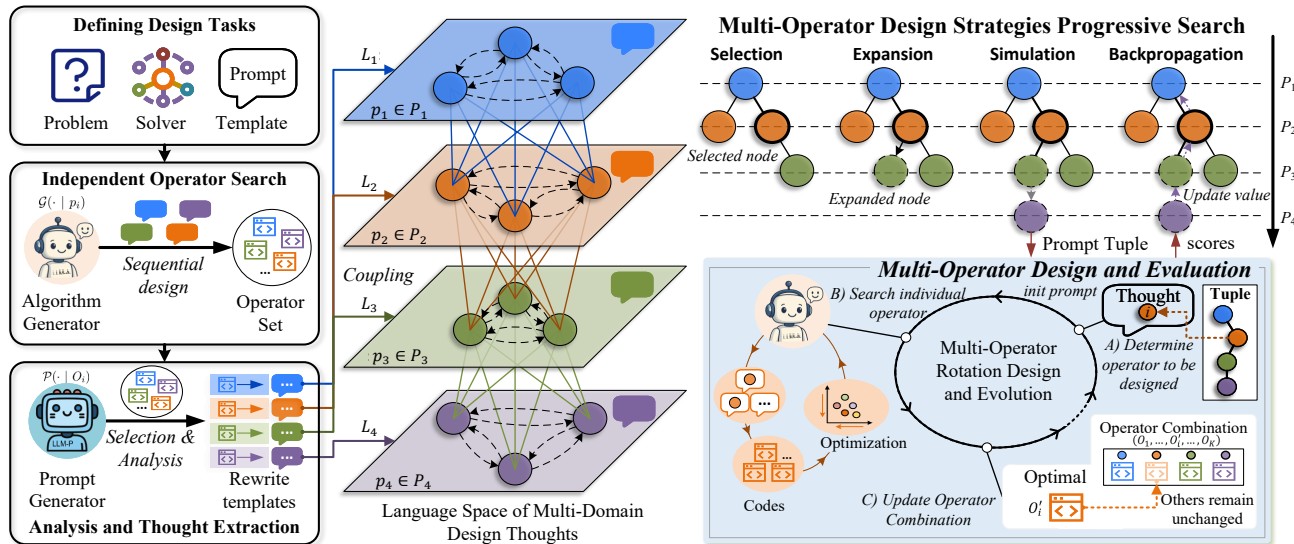

*Figure 2.* The E2OC framework. **Left:** Warm-start stage, where operator sets are independently designed and improvement suggestions are analyzed for each operator. **Center:** The multi-domain design thought language space, in which prompts generated by different operator design thoughts exhibit complex coupling relationships. **Right:** MCTS-based branch selection and expansion are employed to explore promising design strategies, while operator rotation evolution is used to reinforce dominant search paths.

uator $Eva(\cdot)$. It enables rapid evaluation of operators in the instance set $D$ and computes the performance $\bar{F}^N$ as a score $fit$ for the candidate operator combination set $OS$. It will be referenced for operator ranking and elite operator filtering.

**Step 3:** The prompt generator $\mathcal{P}(\cdot)$ analyzes elite operators for dominance and incorporates them as code templates into the initialized prompt storage $PS$, highlighting potential improvements and constraints for future iterations.

Through the above independent evolution, E2OC establishes an interpretable a prior design knowledge surface and a robust code family for each operator, providing reusable design thoughts and code templates support for subsequent strategy search and coupled exploration.

### 3.3. Language Space of Multi-Domain Thoughts

Operator design thoughts correspond to semantic-level representations of operators. Each thought defines a decision paradigm or an improvement direction, and their interactions collectively determine the overall performance. Owing to structural and functional differences, design thoughts for different operators reside in distinct knowledge spaces. Their relationships can be categorized into two types:

**Internal relationships** refer to topological associations among different design thoughts of the same operator. They characterize the relative strengths, weaknesses, and inheritance relationships of alternative implementations and can be evaluated by keeping other operators fixed. **External relationships** denote cross-domain dependencies between

design thoughts of different operators. These include complementary, conflicting, or mutually exclusive effects arising from distinct functional roles. Properly coordinating these design thoughts is crucial for achieving global optimization.

Since design thoughts are expressed in language and knowledge rather than numerical variables, their coupling relationships cannot be directly optimized using conventional numerical methods.

### 3.4. Progressive Design Strategy Search

To address this challenge, we employ a progressive MCTS-based mechanism to explore synergistic paths within the multi-domain design thought space. By balancing exploration and exploitation, this approach identifies effective strategies to guide downstream operator implementation. Here, the design space is modeled as a tree where states of the nodes represent the design thoughts in different domains, and edges denote feasible transitions between them. For a node $j$, the state maintains statistical information including the accumulated performance score $sco_j$ and the visit count $vs_j$. Let $p$ denote its parent node. Node selection is guided by the Upper Confidence Bound (UCB) criterion:

$$UCB_j = \frac{sco_j}{vs_j} + c\sqrt{\frac{\ln(vs_p + 1)}{vs_j}}, \qquad (6)$$

where the first term captures empirical performance, while the second promotes exploration of less-visited paths, with the intensity controlled by $c$ (defaulting to $\sqrt{2}$). Following the standard MCTS procedure summarized in Algorithm 2, each iteration executes four key stages:

**Selection.** Starting from the root node, child nodes are recursively selected according to the UCB criterion, allowing the search to focus on high-potential design paths while preserving exploration of alternative strategies.

**Expansion.** If the selected node has not reached the predefined number $K$ of operators, it is expanded by appending new operator design thoughts sampled from the feasible prompt storage $PS$, extending the current state.

**Simulation.** When the length of the operator thoughts reaches $K$, a multi-operator rotation design and evaluation is performed. Otherwise, additional thoughts are randomly sampled to complete the design state, enabling stochastic rollout and approximate evaluation.

**Backpropagation.** After simulation, the obtained performance feedback is propagated along the search path. The accumulated score $sco_j$ of each visited node is updated using the fitness value $fit_j$, reinforcing effective design strategies and guiding subsequent search decisions.

After each external iteration, MCTS selects the operator combination $\boldsymbol{O}_{best}$ (output $\boldsymbol{O}^*$ when the iteration ends) and prompt tuple $\boldsymbol{p}_{best}$ with the highest score as the best.

### 3.5. Operator Rotation Evolution

The operator rotation mechanism performs multi-operator design and evaluation guided by a design strategy, using an algorithm generator integrated with LLM. During rotation, the operator combination is progressively updated by replacing individual operators and evaluating their impact on overall performance.

Initially, a evaluator $Eva(\cdot)$ assesses the performance of the initial operator combination $\boldsymbol{O}_1$ on the instance set $D$, yielding an initial fitness value $fit$. During each inner iteration, for a given operator $O_i$, its design prompt $p_i$ is retrieved from the prompt tuple $\boldsymbol{p}$, and a candidate operator set $OS_i$ is generated accordingly. The candidate operator $O'_i$ with the highest fitness is selected to replace $O_i \in OS_i$, producing an updated operator combination $\boldsymbol{O}'$ (see Algorithm 3).

The updated combination $\boldsymbol{O}'$ is then evaluated to obtain a new fitness value $fit'$. If $fit'$ exceeds the current best fitness, both the optimal operator combination $\boldsymbol{O}_{best}$ and its fitness are updated. Through repeated inner iterations and operator rotations, the operator combination is progressively refined, resulting in a high-performing operator set and its associated design prompts.

## 4. Experiments

### 4.1. Experimental Setting

**Benchmarks and Datasets.** The proposed E2OC is evaluated on two classical MCOPs: the Multi-objective Flexible

Job Shop Scheduling Problem (FJSP) (Dauzère-Pérès et al., 2024) and the Traveling Salesman Problem (TSP) (Lust & Teghem, 2010b). Both problems are investigated in both bi-objective and tri-objective settings, as detailed in Appendix F and G.

- **FJSP:** Experiments are conducted on the Brandimarte benchmark set (Brandimarte, 1993), which consists of 15 instances of varied scale and complexity, including the highly constrained mk15 instance. The optimization objectives include the minimization of makespan, total machine load, and maximum machine load.

- **TSP:** Following the $M$-objective formulation (Chen et al., 2023), each instance is defined by $M$ distinct two-dimensional coordinate sets for $k$ nodes. A candidate solution is evaluated by $M$ objectives, where the $m$-th objective represents the total tour length calculated within the $m$-th coordinate space. We evaluate performance across problem scales $k \in \{20, 50, 100\}$.

**Hyperparameters.** E2OC is designed offline (similar to offline training) to obtain high-quality multi-operator combinations before online evaluation. The control parameters of E2OC are configured as follows: the external iteration count, intermediate iteration number for operator alternation, and inner iteration number for the algorithm generator are set to 30, 5, and 10, respectively, with a population size of 10. After experimental validation of different models and parameters(see **Appendix**), deepseek-chat is selected as the large language model, with the initial number of newly generated prompts set to 3. The computational resources used in the evaluation process are larger than training.

Detailed parameter specifications of MOEAs for different problems and the configurations of different AHD method are presented in Appendix I.3.

**Baselines.** We compare with several MOEAs using classical operators, advanced AHD methods and co-design frameworks: (1) MOEAs include: a) the dominance-relation-based **NSGA-II** (Deb et al., 2002) and **NSGA-III** (Deb & Jain, 2014) which represent one of the mainstream solutions for MCOPs, and b) the decomposition-based **MOEA/D** (Qingfu Zhang & Hui Li, 2007). These methods employ commonly used crossover, mutation, and neighborhood search operators, as described in **Appendix** F.2 and G.2. (2) Advanced LLM-based single-heuristic design methods: **Random** (Zhang et al., 2024a), **FunSearch** (Romera-Paredes et al., 2024), **EoH** (Liu et al., 2024a), **ReEvo** (Ye et al., 2024) and **MCTS-AHD** (Zheng et al., 2025a). All experiments are conducted on LLM4AD platform (Liu et al., 2024b). (3) Different multi-heuristic co-design frameworks, such as those based on Coordinate-Descent (**CD**), Upper-Confidence-Bound(**UCB**), **LLM**, **MCTS** and their variants,

are described in detail in **Appendix** I.2. All AHD methods are configured to use the same evaluation budget.

**Metrics.** MOEAs are evaluated with Hypervolume (HV) and Inverted Generational Distance (IGD). Results are reported as means over multiple independent runs. For automatic design methods, we assess performance with Relative Improvement (RI), code accuracy, computational cost, and quality cost, highlighting best values in bold.

## 4.2. Main results

**Comparison with Expert Design.** This section compares operator combinations generated by E2OC with expert-designed combinations on multi-objective FJSP and TSP instances in NSGA-II, NSGA-III and MOEA/D. The operator setting details are provided in Appendix F.2 and G.2. Instances are split into training, testing, and all sets. All algorithms are independently executed five times, and the average HV, IGD, and RI are Summarized in Table 1.

*Bi-objective FJSP & TSP.* On bi-objective problems, E2OC consistently outperforms expert-designed operators. Although NSGA-II with expert operators initially performs best, E2OC improves its HV by 22.00% (FJSP) and 14.00% (TSP), and even improved the MOEA/D (TSP) by 16.92%. Across all algorithms, E2OC delivers at least a 10% improvement, indicating that the evolved operators expand the coverage of high-quality solutions in the objective space.

*Tri-objective FJSP & TSP.* On tri-objective problems, performance gains remain significant, albeit slightly lower. With equal evaluation budgets, E2OC improves HV by 17.36% in tri-objective FJSP under NSGA-II, surpassing human-designed paradigms. Notably, E2OC achieves the highest average improvement across all testing instances, demonstrating strong generalization and robustness without overfitting to the training data.

**Comparison with LLM-based AHD Methods** Table 6 compares existing LLM-based AHD methods and multi-heuristic design frameworks on the Bi-FJSP. All methods design operators for NSGA-II starting from identical initial combinations, under the same evaluation budget. Details of the methods are provided in **Appendix** I.2.

*Single-heuristic design.* In single-heuristic design methods, each operator is designed sequentially with equal budget. On the training instance, FunSearch achieves the best mean HV (0.1712) over five independent runs. On testing and all instances, MCTS-AHD performs best, highlighting the effectiveness of MCTS in such co-design scenarios. E2OC consistently outperforms all methods across all instances, reaching the HV of 0.2435. This demonstrates that co-evolving design strategies and codes yields greater advantages than sequentially optimizing operators independently.

*Table 1.* Comparison with expert-designed MOEAs. The mk15 (FJSP) and 100-node TSP instances are used for training, and all other instances form the testing set. Each method is run 5 times, and the mean HV and IGD are reported. RI indicates the relative improvement in HV over the baseline.

| Bi-FJSP | | All instances | | Train instance | | Test instances | | RI↑ |
|---|---|---|---|---|---|---|---|---|
| | Method | HV↑ | IGD↓ | HV↑ | IGD↓ | HV↑ | IGD↓ | |
| Expert | NSGA-II | 0.1996 | 2.2487 | 0.1515 | 1.2512 | 0.2030 | 2.3199 | - |
| | NSGA-III | 0.1927 | 2.4938 | 0.1470 | 1.3507 | 2.5755 | 0.2217 | - |
| | MOEA/D | 0.1853 | 2.8155 | 0.1493 | 1.2450 | 0.1879 | 2.9276 | - |
| E2OC | NSGA-II | 0.2435 | 1.1579 | 0.1985 | 0.6830 | 0.2467 | 1.1918 | 22.00% |
| | NSGA-III | 0.2182 | 1.6684 | 0.1695 | 0.8806 | 0.2217 | 1.7247 | 13.27% |
| | MOEA/D | 0.2256 | 1.4585 | 0.1763 | 0.7566 | 0.2292 | 1.5086 | 21.78% |

| Bi-TSP | | Bi-TSP20 | | Bi-TSP50 | | Bi-TSP100 | | RI↑ |
|---|---|---|---|---|---|---|---|---|
| | Method | HV↑ | IGD↓ | HV↑ | IGD↓ | HV↑ | IGD↓ | |
| Expert | NSGA-II | 0.3881 | 0.3439 | 0.3484 | 0.3638 | 0.4079 | 0.3340 | - |
| | NSGA-III | 0.3656 | 0.3971 | 0.3244 | 0.4097 | 0.3862 | 0.3908 | - |
| | MOEA/D | 0.3674 | 0.4045 | 0.3293 | 0.4050 | 0.3865 | 0.4042 | - |
| E2OC | NSGA-II | 0.4424 | 0.2257 | 0.4281 | 0.2206 | 0.4495 | 0.2282 | 14.00% |
| | NSGA-III | 0.4125 | 0.2957 | 0.3854 | 0.2851 | 0.4260 | 0.3009 | 12.81% |
| | MOEA/D | 0.4296 | 0.2279 | 0.4060 | 0.2336 | 0.4414 | 0.2251 | 16.92% |

| Tri-FJSP | | All instances | | Train instance | | Test instances | | RI↑ |
|---|---|---|---|---|---|---|---|---|
| | Method | HV↑ | IGD↓ | HV↑ | IGD↓ | HV↑ | IGD↓ | |
| Expert | NSGA-II | 0.1266 | 1.8398 | 0.0960 | 0.9831 | 0.1287 | 1.9010 | - |
| | NSGA-III | 0.1200 | 2.1074 | 0.0928 | 1.0744 | 0.1220 | 2.1812 | - |
| | MOEA/D | 0.1116 | 2.3652 | 0.0786 | 1.2435 | 0.1139 | 2.4454 | - |
| E2OC | NSGA-II | 0.1485 | 1.1619 | 0.1183 | 0.6368 | 0.1507 | 1.1994 | 17.36% |
| | NSGA-III | 0.1407 | 1.4379 | 0.1111 | 0.7956 | 0.1428 | 1.4838 | 17.24% |
| | MOEA/D | 0.1229 | 1.9366 | 0.0820 | 1.1123 | 0.1258 | 1.9955 | 10.15% |

| Tri-TSP | | Bi-TSP20 | | Bi-TSP50 | | Bi-TSP100 | | RI↑ |
|---|---|---|---|---|---|---|---|---|
| | Method | HV↑ | IGD↓ | HV↑ | IGD↓ | HV↑ | IGD↓ | |
| Expert | NSGA-II | 0.1824 | 0.2235 | 0.1266 | 0.2020 | 0.2104 | 0.2342 | - |
| | NSGA-III | 0.1773 | 0.2251 | 0.1215 | 0.2028 | 0.2052 | 0.2363 | - |
| | MOEA/D | 0.1800 | 0.2221 | 0.1251 | 0.1997 | 0.2074 | 0.2333 | - |
| E2OC | NSGA-II | 0.1939 | 0.2200 | 0.1333 | 0.2075 | 0.2243 | 0.2262 | 6.30% |
| | NSGA-III | 0.1873 | 0.2110 | 0.1292 | 0.1983 | 0.2163 | 0.2173 | 5.63% |
| | MOEA/D | 0.1867 | 0.2052 | 0.1286 | 0.1858 | 0.2157 | 0.2149 | 3.73% |

*Multi-heuristic design.* Among multi-operator co-design frameworks, methods based on CD, UCB, and direct LLM-guided decisions degrade in performance; the purely LLM-driven approach performs worst. This confirms that relying solely on LLMs or expert knowledge to directly control operator combination design in MOEAs is ineffective. Win-UCB achieves the second-best result after E2OC, with an HV of 0.1763 on the training instance, which significantly surpasses all single-heuristic designs, but it underperforms on other instances. E2OC attains the highest average performance across all instances, indicating that its learned strategies generalize robustly to different problem instances rather than overfitting to the training data.

## 4.3. Ablation Studies

The core components of E2OC comprise progressive design-strategy search and operator-rotation evaluation. We perform ablations on these components to examine their respective contributions. In variant **MCTS_OC**, the progressive search of operator combinations is replaced with a single fixed design strategy. In variant **E2OC-SD**, the operator-rotation mechanism is replaced with sequential

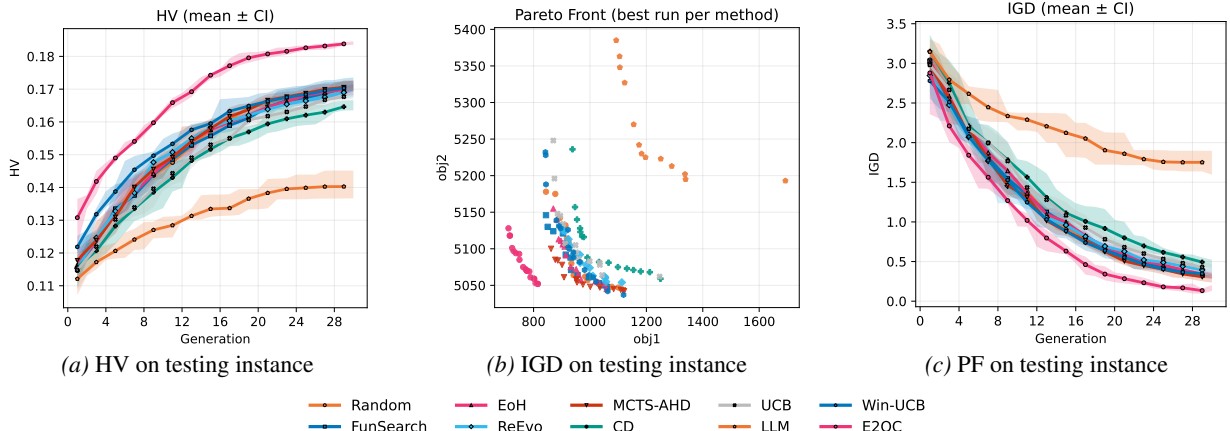

*(a)* HV on testing instance   *(b)* IGD on testing instance   *(c)* PF on testing instance

Random   EoH   MCTS-AHD   UCB   Win-UCB
FunSearch   ReEvo   CD   LLM   E2OC

*Figure 3.* Performance of different AHD methods on BIFJSP testing (mk14) instances.

*Table 2.* Comparison of A·HD methods on Bi-FJSP. All operator combinations are evaluated 5 times independently in NSGA-II, with mean performance reported. Best values are in bold.

| Type | Method | All instaces | | Train instace | | Test instaces | |
|---|---|---|---|---|---|---|---|
| | | HV↑ | IGD↓ | HV↑ | IGD↓ | HV↑ | IGD↓ |
| Single | Random | 0.2263 | 1.4193 | 0.1702 | 0.8546 | 0.2303 | 1.4597 |
| | FunSearch | 0.2265 | 1.4070 | 0.1712 | 0.8386 | 0.2210 | 1.4193 |
| | EoH | 0.2258 | 1.4352 | 0.1694 | 0.8409 | 0.2298 | 1.4777 |
| | ReEvo | 0.2185 | 1.6551 | 0.1669 | 0.8712 | 0.2222 | 1.7110 |
| | MCTS-AHD | 0.2269 | 1.3950 | 0.1709 | 0.8371 | 0.2309 | 1.4348 |
| Multi | CD | 0.2170 | 1.6536 | 0.1630 | 0.9572 | 0.2209 | 1.7033 |
| | UCB | 0.2182 | 1.6300 | 0.1663 | 0.9296 | 0.2219 | 1.6800 |
| | LLM | 0.2148 | 1.8772 | 0.1654 | 0.9424 | 0.2183 | 1.9440 |
| | Win-UCB | 0.2256 | 1.4619 | 0.1763 | 0.7566 | 0.2292 | 1.5123 |
| | E2OC | **0.2435** | **1.1423** | **0.1985** | **0.6830** | **0.2467** | **1.1751** |

*Table 3.* Ablation study with different design baselines. All designs are evaluated 5 times in NSGA-II; mean performance is reported, with best values in bold.

| Method | All instancess | | Train instances | | Test instancess | |
|---|---|---|---|---|---|---|
| | HV↑ | IGD↓ | HV↑ | IGD↓ | HV↑ | IGD↓ |
| MCTS_OC | 0.2085 | 1.8893 | 0.1583 | 1.1519 | 0.2121 | 1.9420 |
| E2OC-SD | 0.2187 | 1.8893 | 0.1713 | 0.8272 | 0.2221 | 1.6484 |
| E2OC[EoH] | **0.2435** | **0.2264** | **0.1985** | **0.7349** | **0.2467** | **1.1752** |
| E2OC[FunSearch] | 0.2264 | 1.4115 | 0.1754 | 0.7908 | 0.2300 | 1.4558 |
| E2OC[MCTS-AHD] | 0.2269 | 1.3961 | 0.1785 | 0.7924 | 0.2303 | 1.4393 |
| E2OC[ReEvo] | 0.2213 | 1.5289 | 0.1719 | 0.8340 | 0.2248 | 1.5786 |

operator design, where the evaluation budget is uniformly allocated across operators and design proceeds sequentially. We further test several advanced AHD methods to verify compatibility with different operator designers. The HV and IGD performance across instance are reported in Table 3.

Removing progressive search causes notable performance degradation in MCTS_OC, indicating that under a fixed evaluation budget, the ability to switch design strategies is more effective than sequential operator design with a fixed strategy. Moreover, compared to sequential independent design, the operator rotation mechanism finds local optima more efficiently in complex algorithm spaces, which significantly accelerates the evaluation of operator strategies. The choice of operator designer also affects overall performance, with EoH achieving the best results.

## 5. Discussion and Future Works

### 5.1. Discussion

**Different LLMs** The performance of E2OC and other LLM-based AHD methods depends significantly on the underlying LLM, which influences the quality of generated design thoughts and operator code. We evaluate six representative LLMs from the DeepSeek, GPT, Qwen, and Gemini families on the NSGA-II operator design task for Bi-FJSP under consistent experimental conditions. Search performance, evaluation performance, and computational cost are summarized in Table 4. Results indicate that stronger general-purpose LLMs do not consistently achieve better performance in improvement summarization or operator design on E2OC . Although gpt-4.1-mini attains the best search performance on some instances, deepseek-chat offers a more favorable trade-off overall, with competitive performance, lower evaluation cost, and a higher quality-cost ratio. Therefore, deepseek-chat is selected as the default backbone model in subsequent experiments.

**Different MCTS variants** MCTS supports effective reuse of branching information and offers a structured framework for modeling dependencies among multiple operators. We compare four MCTS variants for multi-operator co-design according to state representation and expansion mechanisms: MCTS_OC progressively searches operator combinations; MCTS_Tuple progressively searches design strategies with node states as design-thought tuples; MCTS_Sample progressively samples and searches design thoughts during expansion; and E2OC searches a warm-start-built design

*Table 4.* Comparison of different LLMs. The $^*$ denotes open-source models, which are less expensive for local deployment. Ratio denotes the quality-cost ratio, defined as performance improvement over the baseline divided by cost. Shading marks the most cost-effective model.

| LLM | Search performance | | | Evaluation | | Expenditure | | |
|---|---|---|---|---|---|---|---|---|
| | ValidR↑ | Mean↑ | Range↑ | HV↑ | IGD↓ | Tok.(M) | Cost($)↓ | Ratio↓ |
| deepseek-chat | 99.76% | 0.1485 | **0.168** | **0.2271** | 1.5437 | 3.34 | **1.14** | 41.53 |
| gpt-4.1-mini | 99.97% | **0.1502** | 0.043 | 0.2266 | **1.4234** | 3.41 | 1.22 | 45.23 |
| gpt-4o-mini | **100.00%** | 0.1458 | 0.161 | 0.2211 | 1.5840 | **2.44** | 2.23 | 103.29 |
| qw3-8b$^*$ | 99.93% | 0.1475 | 0.163 | 0.2258 | 1.4556 | 2.83 | 3.19 | 121.99 |
| qw3-30b-A3b$^*$ | 99.93% | 0.1473 | 0.165 | 0.2244 | 1.4997 | 5.39 | 13.56 | 222.87 |
| gemini-2.5-pro | 99.90% | 0.1482 | 0.163 | 0.2223 | 1.5745 | 14.35 | 117.84 | 5196.28 |

thought space (details in **Appendix** E).

For fairness, MCTS_OC and MCTS_Tuple impose no restriction on LLM sampling, while MCTS_Sample and E2OC limit the sampled design thoughts per operator to $AP$, with E2OC pre-constructing the design strategy space. Experimental results are summarized in Table 5. Under a fixed evaluation budget, E2OC attains the best performance across all instance sets, achieving HV values of 0.1985 (training), 0.2467 (testing), and 0.2435 (all). These results demonstrate that searching in a fixed, structured design-thought space identifies high-quality design strategies more efficiently than dynamically sampling thoughts during tree expansion.

*Table 5.* Comparison of MCTS variants and validation of continuous optimization. Gray highlighting indicates the best single-run performance; bold denotes the global optimum.

| Method | All instances | | Train instances | | Test instances | |
|---|---|---|---|---|---|---|
| | HV↑ | IGD↓ | HV↑ | IGD↓ | HV↑ | IGD↓ |
| MCTS_OC | 0.2085 | 2.3828 | 0.1583 | 2.0977 | 0.2121 | 2.4031 |
| MCTS_Tuple | 0.2186 | 2.0888 | 0.1655 | 1.9628 | 0.2224 | 2.0978 |
| MCTS_Sample | 0.2181 | 2.1088 | 0.1633 | 1.9905 | 0.2220 | 2.1172 |
| E2OC | 0.2435 | 1.7749 | 0.1985 | 1.1188 | 0.2467 | 1.8217 |
| E2OC′ | 0.2454 | 1.6617 | 0.1986 | 1.1095 | 0.2487 | 1.7012 |
| E2OC″ | **0.2475** | **1.5453** | **0.1999** | **1.0238** | **0.2509** | **1.5826** |

**Continuous Optimization Analysis**  The design thought space of E2OC is initialized from an elite operator set obtained during the warm-start phase, allowing progressive discovery of strategies and operator combinations that surpass expert-designed baselines. To examine its potential for sustained optimization, we conduct three consecutive E2OC runs on the NSGA-II operators design task for Bi-FJSP, where the output strategies and operators of each run are reused as inputs for the next.

The results show consistent performance improvements in both the second E2OC′ and third E2OC″ runs compared with the previous ones in Table 5, demonstrating that initializing E2OC with increasingly stronger design strategies and operator sets further enhances performance. This iterative reconstruction of the design thought space endows E2OC

with clear continuous optimization capability.

**Evolutionary Process Visualization**  Understanding the progressive search process of E2OC's semantic-level design strategies is essential for exploring the complex coupling relationships among multiple operators or their corresponding design thoughts. The evolution process of the crossover and mutation operators for processes and machines in NSGA-II for solving BIFJSP is optimized by E2OC, as shown in Figure 4, and the design thoughts and specific suggestions are summarized in Table 13 and Figure 15.

During the warm-start phase, all operators (colored circles) generate distinct design thoughts (numeric indices). MCTS then explores various combinations of these thoughts to identify the most effective design strategy. Over 30 iterations, these thoughts form composite strategies. The algorithm generator employs an operator rotation evolution mechanism to design operator combinations, integrating them into NSGA-II for three independent runs. The average Hypervolume (HV) serves as the performance score. Notably, generations 11 and 24 share the same strategy (3,1,0,1) but yield different scores, confirming that a single evaluation is insufficient and requires iterative sampling.

MCTS continuously balances exploration and exploitation, prioritizing branches with higher scores. Specifically, (0,0,0,0) denotes the manual initial design thoughts (Figure 15), while the green curve tracks E2OC's best-discovered score. A local optimum is reached at generation 4 (HV 175.6, up from 169.4 at generation 3). Subsequent variations around operation mutation (index 0) and machine crossover (index 2) yield no improvement. Performance rises to 176.8 at generation 10, and a superior strategy (1,1,1,1) emerges at generation 12, achieving a peak HV of 181.9 and significantly outperforming the baseline.

For LLM-based algorithm design, input text directly shapes the generation trajectory. These four design thoughts form a complete, complementary design space covering parameter adaptation, strategy hybridization, information utilization, and goal orientation. Effective design spaces must balance dimensional coverage and synergy; focusing on a single dimension limits generalization. By searching within this validated space, the LLM is guided to generate high-performance algorithms.

### 5.2. Future Works

Although LLM-based AHD has recently gained attention, it remains at an early stage of development. Existing studies indicate substantial potential in automatic algorithm design, warranting deeper and more systematic investigation.

**Semantic-level Optimization**  LLMs introduce an optimization paradigm that operates in semantic spaces rather

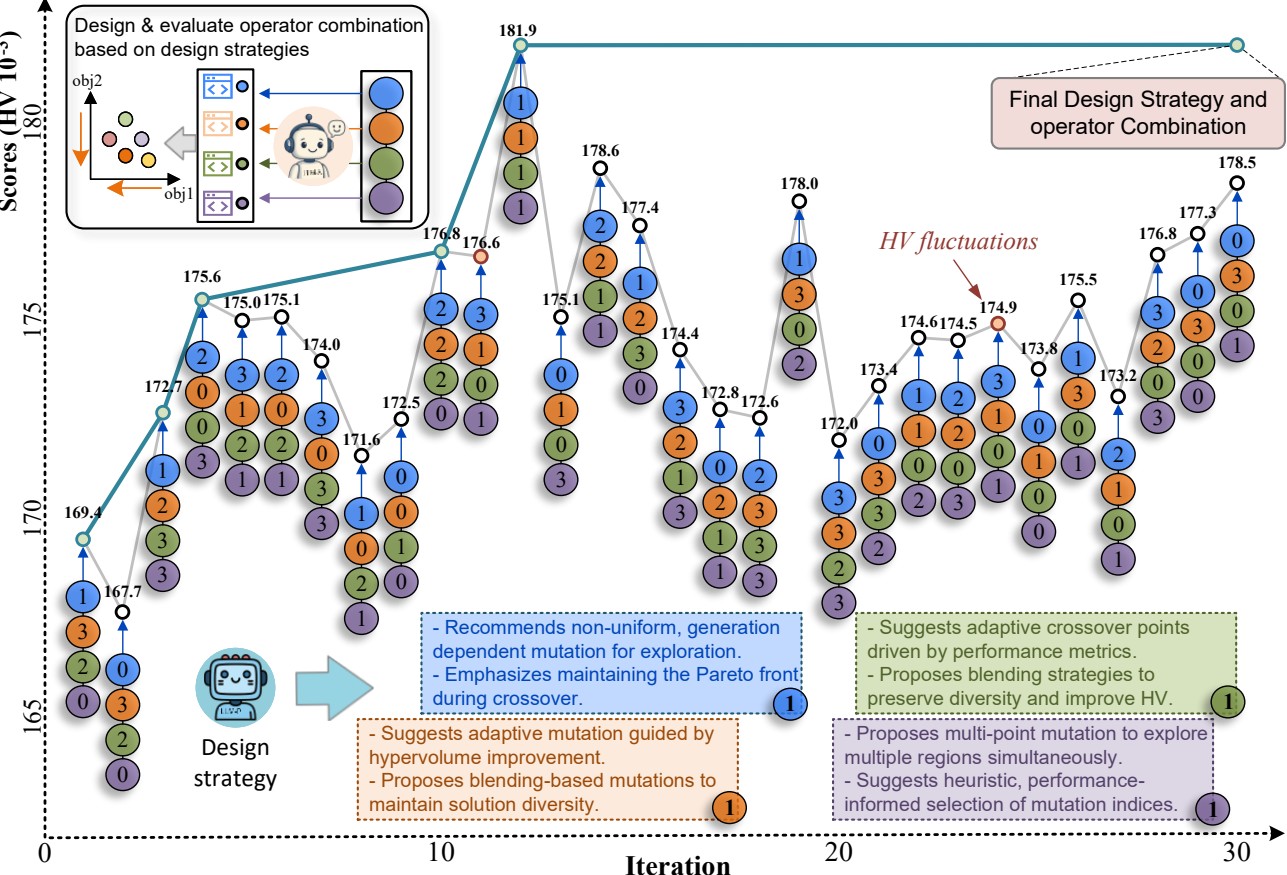

*Figure 4.* The evolution process of E2OC to design the operator combinations in NSGA-II for solving Bi-FJSP. The optimal design strategy is searched through the iteration of MCTS. Different colored balls denote different FJSP operators, and the number indicates the corresponding design thought index.

than purely numerical or combinatorial domains. Future research should focus on developing principled formulations and tools for semantic-level optimization, where language and knowledge representations define the search space and optimization dynamics.

**Human–AI Co-design** The design strategies in E2OC provide continuous guidance for heuristic evolution and improve interpretability. However, practical deployment under complex constraints requires effective integration of expert knowledge. Future work could explore interactive optimization frameworks that incorporate human preferences or expert evaluations to guide and accelerate algorithm design.

**Autonomous Algorithmic System Evolution** Current LLM-based AHD frameworks have demonstrated promise but often rely on limited supervision and relatively simple algorithm structures. A key direction is to exploit LLMs' self-reflection and multi-agent coordination capabilities to support autonomous, iterative evolution of algorithmic systems, which calls for systematic modeling of algorithm

representations and their dynamic evolution mechanisms.

## 6. Conclusion

This paper investigates the automated evolution of the interdependent operators in MOEAs. We propose an E2OC framework that co-evolves design strategies with executable codes of operator combination. Progressive search based on Monte Carlo trees is employed to explore combinations of different operator design thoughts. The optimal operator combination is systematically searched and determined through operator rotation evolution, and supports the integration of mainstream AHD methods as the underlying algorithm generator. We evaluate E2OC on both bi-objective and tri-objective FJSP and TSP. The experimental results show that E2OC consistently outperforms mature human-designed operators across multiple MOEAs. By explicitly modeling and exploring interdependencies among operators, E2OC also achieves superior performance compared to state-of-the-art AHD methods. The source code can be found in https://github.com/jhqiu1/E2OC.

## Acknowledgements

The work described in this paper was supported by the Research Grants Council of the Hong Kong Special Administrative Region, China (GRF Project No. CityU11217325), and the Natural Science Foundation of China (Project No: 62276223).

## Impact Statement

This paper presents work whose goal is to advance the field of Machine Learning. There are many potential societal consequences of our work, none of which we feel must be specifically highlighted here.

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

## A. Reproducibility Statement

The formulation and Markov Decision Process of multi-operator optimization are presented in Section 2 and Section C, respectively. The technical foundation of the E2OC framework is detailed in Section 3, with further explanations provided in Appendix D, and prompts are detailed in Appendix D.4. To ensure full reproducibility, all essential experimental components are documented in the supplementary material. This material includes descriptions of the different problems and operator settings (Appendix F and G), hyperparameters settings (Appendix I.3), details of the comparison methods (Appendix E and I.2), and additional results (Appendix J). In addition, the supplementary material contains source code to enable direct replication of all reported experiments.

## B. Related Works

### B.1. Automatic Heuristic Design (AHD)

**General AHD Method.**    Automatic heuristic design, also known as hyper-heuristics (Burke et al., 2013; Stützle & López-Ibáñez, 2019), aims to automatically generate, select, or adapt heuristics for complex optimization problems, reducing reliance on expert knowledge and enhancing cross-domain applicability (Burke et al., 2019; Akiba et al., 2019).

Genetic Programming (GP)-based methods form a major paradigm within AHD. They encode heuristics as tree-structured programs or executable code and evolve high-performing rules through genetic operators such as crossover, mutation, and selection within a defined search space (Jia et al., 2022; Mei et al., 2022). Genetic Programming (GP)-based AHD has been applied to domains such as production scheduling (Nguyen et al., 2017; Zhang et al., 2023) and path planning (Jia et al., 2022; Ardeh et al., 2021), demonstrating its ability to discover complex heuristic strategies, including priority rules in scheduling and composite heuristics for combinatorial problems.

Despite their effectiveness, these methods rely on manually designed genetic operators and fixed terminal and function sets (Pillay & Qu, 2018; O'Neill et al., 2010). The resulting heuristics often have complex, hard-to-interpret structures, which limits scalability and practical applicability.

**LLM-driven AHD.**    LLMs have enabled a new paradigm for AHD, in which heuristic construction is guided by language-based generative models (Liu et al., 2026; Zhang et al., 2024b). In this setting, AHD is often formulated as an evolutionary program search, where LLMs generate and modify algorithmic code within an evolutionary optimization framework. These approaches have shown strong empirical performance in optimization (Liu et al., 2024a; Ye et al., 2024; van Stein & Bäck, 2024; Ye et al., 2025; Dat et al., 2025; Yao et al., 2025; Li et al., 2025; Qiu et al., 2026), mathematical discovery (Romera-Paredes et al., 2024; Novikov et al.), black-box optimization (Ma et al., 2025; Xu et al., 2025) and machine learning (Mo et al., 2025).

FunSearch provides a baseline by using an island-based evolutionary framework with a single prompting strategy for LLM-driven code optimization (Romera-Paredes et al., 2024). Building on this, the EoH jointly evolves heuristic ideas and executable code (Liu et al., 2024a), introducing multiple prompting strategies to enhance diversity and exploration. MEOH further incorporates dominance-based selection on objective vectors (Yao et al., 2025), enabling efficient multi-objective heuristic design within the same LLM-driven evolutionary setting.

Beyond evolutionary approaches, alternative search paradigms have been explored. Monte Carlo Tree Search (MCTS) guides LLM-based heuristic synthesis through structured exploration (Zheng et al., 2025a; Kiet et al., 2025), while neighborhood search improves sample efficiency and local refinement (Xie et al., 2025). Collectively, these LLM-driven methods reduce dependence on manually predefined symbol sets and operate in semantically richer spaces, enabling more flexible and expressive heuristic synthesis.

### B.2. Operator Selection and Design in Evolutionary Algorithms

The effectiveness of evolutionary algorithms largely depends on the employed operators. Existing research on operator selection and design can be broadly categorized into two directions: adaptive operator selection based on search feedback, and the design of problem-informed operators that exploit domain-specific structures (Pei et al., 2025).

**Operator Selection.**    Operator selection is commonly studied as an online decision-making problem, where operator usage is adapted according to the current search state and historical performance. In many studies, this process is further

formulated within the frameworks of adaptive parameter control (Huang et al., 2019) or selection hyper-heuristics (Drake et al., 2020), in which each candidate operator is treated as an alternative parameter configuration or heuristic strategy to be selected during the evolutionary search. Methods such as adaptive large neighborhood search (Mara et al., 2022; Tang et al., 2009) and reinforcement learning model (d O Costa et al., 2020; Zhao et al., 2023) selection as a policy mapping states to operator choices, with feedback derived from fitness gain or diversity preservation. This allows operator distributions to adapt dynamically, reducing reliance on fixed schedules or expert heuristics.

**Operator Design.**    Operator design is undergoing a transition from expert-crafted operators (Helsgaun, 2000; Lan et al., 2021) tailored to specific problems and objectives toward automated design approaches. Early methods based on GP evolve operator components and compositions within predefined structural spaces (Hong et al., 2018). More recently, LLM–based approaches have demonstrated the ability to automatically generate and explore operator structures via natural language specifications or code synthesis (Liao et al., 2025), providing a more expressive and flexible design space.

**Operator Combination in Multi-Objective EAs.**    In multi-objective EAs (MOEAs), operators must balance convergence toward the Pareto front with preservation of solution diversity (Li et al., 2022). Single operators often bias the search toward specific regions. To mitigate this, methods combine multiple mutation or recombination operators with complementary behaviors and manage their usage through cooperative or competitive strategies (Chen et al., 2025; Li et al., 2024a). The challenge lies in designing operator sets and selection mechanisms that adapt to dynamic Pareto fronts while accounting for inter-operator interactions rather than evaluating operators in isolation.

### B.3. MCTS with LLM for Structured Reasoning and Decision-Making

Monte Carlo Tree Search (MCTS) is a simulation-based heuristic search for large, structured decision spaces, providing approximate solutions when exhaustive search is infeasible (Coulom, 2007). It operates through a cycle of selection, expansion, simulation, and backpropagation, balancing exploration of uncertain paths with exploitation of promising ones (Świechowski et al., 2023). Beyond passive evaluation, MCTS can actively guide decision-making by coordinating with deep neural networks (Silver et al., 2016; Fu et al., 2021), demonstrating effectiveness in complex spaces.

**MCTS with LLM.**    Recent work integrates MCTS with LLMs for structured reasoning and algorithm design (Zheng et al., 2025a; Kiet et al., 2025). In one approach, LLMs generate reasoning steps while MCTS evaluates and selects promising paths, as in the Tree of Thoughts framework, enabling self-correcting multi-step reasoning (Wei et al., 2022; Yao et al., 2023). In another, MCTS-AHD (Zheng et al., 2025a), LLMs produce candidate heuristic configurations and MCTS manages search via progressive expansion in large program spaces. MOTIF extends this to multi-strategy co-design, facilitating turn-based optimization between two LLM agents (Kiet et al., 2025).

These methods combine LLMs' generative flexibility with MCTS's selective control. LLMs provide diverse candidate solutions, while MCTS guides efficient search through hierarchical selection and backpropagation. This integration enhances structured reasoning and offers a scalable framework for complex decision problems, transforming MCTS into a dynamic controller of LLM outputs.

### B.4. Reflective Prompting

Reflective prompting enables LLMs to iteratively generate, evaluate, and revise outputs, forming a generate-reflect-revise loop to improve quality (Shinn et al., 2023). In automated algorithm design, ReEvo embeds reflection into evolutionary search (Ye et al., 2024), allowing LLMs to compare algorithm variants and extract insights to guide subsequent search. LLM4EO applies reflective prompting in operator design to identify patterns from successful instances and enable knowledge reuse (Liao et al., 2025). By integrating reflection with optimization, these methods support multi-round self-assessment and learning from past errors, improving efficiency and generalization in complex design tasks.

## C. Markov Decision Process for Multi-Operator Optimization

The interdependent multi-operator evolution process is inherently dynamic rather than a static multi-variable optimization problem. Modifying one operator changes the generation distributions of other operators, continuously reshaping the overall search landscape during evolution. This process can be formulated as a Markov Decision Process (MDP), represented by the tuple $(S, A, P, R)$, where the optimization proceeds iteratively over time steps $t \in T$.

**State Space.** At iteration $t$, the state $s_t \in S$ represents the current operator combination and its associated prompt information, defined as:

$$s_t = (O_{1,t}, O_{2,t}, \ldots, O_{K,t} \mid \boldsymbol{P}_t), \tag{7}$$

where $O_{i,t}$ denotes the $i$-th operator at step $t$, and $\boldsymbol{P}_t = (p_{1,t}, p_{2,t}, \ldots, p_{K,t})$ represents the corresponding prompt tuple, with $p_{i,t}$ being the prompt template used for the generation of $O_{i,t}$.

**Action Space.** During the operator combination evolution, each action determines which operator $i$ should be evolved and whether its corresponding prompt should be regenerated. Let $w_i \in \{0, 1\}$ denote the binary decision to rewrite the prompt $p_i$. Thus, each action $a_t = (i, w_i)$ specifies both the target operator and its prompt update decision. Changing a prompt directly affects the generation distribution of the LLM-based algorithm constructor $\mathcal{G}_i(\cdot \mid p_{i,t})$.

**Reward Function.** The reward function guides the evolution of the operator combination toward the optimization objective. Based on scalarized evaluation, the reward improvement at step $t + 1$ is defined as:

$$R(s_t, a_t, s_{t+1}) = \bar{F}^N(d \mid \boldsymbol{O}') - \bar{F}^N(d \mid \boldsymbol{O}), \tag{8}$$

where $\boldsymbol{O}$ and $\boldsymbol{O}'$ represent the operator combinations before and after applying action $a_t$, respectively. A positive reward indicates an improvement in the scalarized performance metric, guiding the evolutionary search toward more effective operator configurations.

Given a total computational budget $T$, the overall optimization objective is to maximize the expected cumulative reward:

$$\max_{\boldsymbol{O} \in \mathbf{S}} \mathbb{E}_{d \sim \mathcal{D}} \left[ \sum_{t=0}^{T-1} R(s_t, a_t, s_{t+1}) \right], a_t \sim \pi(\cdot \mid s_t), \tag{9}$$

where $\pi(a_t \mid s_t)$ denotes the decision policy over actions (e.g., operator selection and prompt rewriting), and $\boldsymbol{O}$ denotes the operator set and their corresponding prompts to be evolved by the prompt generator $\mathcal{G}(\cdot \mid \mathcal{P}(\cdot))$. The expectation is taken over problem instances $d \in \mathcal{D}$ and the stochasticity introduced by the solver $\mathcal{S}$, the generator $\mathcal{G}$, and the evaluation process. This formulation captures the adaptive evolution of operator and prompt configurations within the given computational budget $T$, aiming to maximize the cumulative improvement in scalarized multi-objective performance.

**Transition Probability.** Given a state $s_t$ and an action $a_t$ at decision step $t$, the generator $\mathcal{G}_i(\cdot \mid p_{i,t})$ samples $q$ operator codes $\{c_{i,1}, c_{i,2}, \ldots, c_{i,q}\}$. Fixing all other operators, $q$ new combinations $\{s'_1, s'_2, \ldots, s'_q\}$ are evaluated. The next state $s_{t+1}$ is selected according to the transition probability $P(s_{t+1} \mid s_t, a_t)$, which depends on the performance of the sampled combinations.

## D. Methodology Details

This section provides a comprehensive description of the algorithmic details and the prompts employed. The implementation of E2OC co-evolves design strategies and executable codes through three key stages: warm-start, progressive search, and operator rotation evolution.

### D.1. Warm-start

Before the warm-start phase, the operator combinations to be searched and the corresponding code templates (which serve as key components of the prompts for the algorithm generator) should be pre-specified to form the initial prompt tuple. Additionally, the instance set $D$ for algorithm design and the number of newly introduced prompts need to be determined. This parameters serves as input to E2OC, which then constructs an initial multi-operator set and extracts the corresponding design thoughts.

The detailed procedure is outlined in Algorithm 1. First, an initial population is generated for each operator based on the provided prompt templates. This population is evaluated using a multi-objective evaluator composed of MOEAs and the target optimization problem. Next, a prompt generator extracts design thoughts from the higher-performing operators. It should be noted that each prompt incorporates a different design thought and a fixed structural components. Therefore, in order to make it easier for the reader to understand, this paper directly uses prompts to represent design thoughts. For each

---

**Algorithm 1** Warm-start for Design Thought Extraction.

---

1: **Input:** Initial operator combination $\boldsymbol{O}_1$ and prompt tuple $P_1$ at iteration step $t = 1$; Instance set $D$; Number of initial added prompt $AP$; Candidate operator combination set $OS$; Number $K$ of operators to be evolved.
2: **Output:** Initial operator design prompt storage $PS$.
3: Initialize multi-objective optimization evaluator $Eva$;
4: **for** $i = 1, \ldots, K$ **do**
5: $\quad p_i \leftarrow P_{i,1}$; # $P_{i,1} \in P_1$
6: $\quad$ Initialize algorithm generator $\mathcal{G}_i(\cdot \mid p_i)$ of $O_i$;
7: $\quad$ **for** $j = 1, \ldots, ON_{max}$ **do**
8: $\quad\quad O_{i,j} \leftarrow \mathcal{G}_i(\cdot \mid p_i)$;
9: $\quad\quad \boldsymbol{O}'_1 \leftarrow$ Update the operator $i$ in $\boldsymbol{O}_1$ with $O_{i,j}$;
10: $\quad\quad fit_{i,j} \leftarrow Eva(D|\boldsymbol{O}'_1)$;
11: $\quad\quad OS_{i,j} \leftarrow (O_{i,j}, fit_{i,j})$;
12: $\quad$ **end for**
13: **end for**
14: $\widehat{OS} \leftarrow$ Sort by $fit_{i,j}$ and filter invalid operators in $OS$;
15: Initialize prompt storage $PS$;
16: **for** $i = 1, \ldots, K$ **do**
17: $\quad ON_i \leftarrow$ select the the smaller of $AP$ and $\widehat{OS}_i$;
18: $\quad PS_{i,1} \leftarrow P_{i,1}$;
19: $\quad$ **for** $g = 1, \ldots, ON_i - 1$ **do**
20: $\quad\quad O_{i,g+1} \leftarrow \widehat{OS}_{i,g}$;
21: $\quad\quad$ Initialize prompt generator $\mathcal{P}(\cdot \mid O_{i,g+1})$ of $O_i$;
22: $\quad\quad PS_{i,g+1} \leftarrow \mathcal{P}(\cdot \mid O_{i,g+1})$;
23: $\quad$ **end for**
24: **end for**

---

initial operator set, a specialized prompt is constructed to instruct the LLM to summarize valuable design insights from elite operators, resulting in the initial operator design prompt storage $PS$. The multiple design thoughts belonging to different operators in PSform an interdependent language space. A progressive search strategy based on MCTS is then employed to explore design strategies within this space.

### D.2. Progressive Search

After obtaining the prompt storage $PS$ and hyperparameters such as the iteration count, the MCTS-based progressive search process is executed as outlined in Algorithm 2. The search iterates through four core phases: selection, expansion, simulation, and backpropagation. Initially, a root node $\mathcal{N}_0$ is selected, and a design thought from $PS$ is chosen as the initial state. It is important to note that nodes in the MCTS tree can be organized into multiple domains, with each domain representing design strategies for the same operator. Thus, a complete design strategy path is formed by concatenating design thoughts across different domains (i.e., design thoughts for different operators). During the expansion phase, a new child node is created and linked to its parent, corresponding to line 17 in the algorithm. In the simulation phase, the operator-rotation evaluation (corresponding to the `OperatorRotationEvaluation` in Algorithm 2) is performed only when the path length equals the total number of operators. If this condition is not met, a new design thought (i.e., prompt) is randomly sampled for temporary evaluation, corresponding to lines 21–25. Finally, the resulting fitness score $fit_j$ from the evaluation is used to update the scores of the corresponding branches in the Monte Carlo tree.

In each subsequent iteration, a new node is selected based on the scores $fit$ of the existing nodes, and the cycle of expansion, simulation, and backpropagation continues. Through continuous exploration and exploitation of the design thoughts in $PS$, the optimal design strategy $\boldsymbol{P}_{best}$ is identified to guide the generation of multiple operators.

### D.3. Operator Rotation Evolution

For node states $P_j$ that satisfy the length requirement, operator rotation evolution is performed, as outlined in Algorithm 3. Similar to coordinate rotation strategies, the number of rotations is controlled by the Max number of middle iterations. In

---

**Algorithm 2** Progressive Search for Design Strategy.

---

1: **Input:** Max number of outer iterations $N_{outer}$; Number of operators to be evolved $K$; Operator design prompt storage $PS$; Storage of operator combination $\boldsymbol{SO}$.
2: **Output:** Top scoring operator combination $\boldsymbol{O}_{best}$ and prompt combination $\boldsymbol{P}_{best}$.
3: $\mathcal{N}_0 \leftarrow$ root node (empty state);
4: **for** $j = 1, \ldots, N_{outer}$ **do**
5:     *# Selection*
6:     $\mathcal{N}_j \leftarrow \mathcal{N}_0$;
7:     **while** $\mathcal{N}_j$ has child node **do**
8:         $NS \leftarrow$ get the child node set of $\mathcal{N}_j$;
9:         $\mathcal{N}_j \leftarrow$ select the node of highest UCB score in $NS$;
10:     **end while**
11:     *# Expansion*
12:     $i \leftarrow$ the size of state in $\mathcal{N}_j$;
13:     $sta_j \leftarrow$ get the state of $\mathcal{N}_j$;
14:     **if** $i<$ **then**
15:         **for** $p_g \in PS_i$ **do**
16:             $sta_g \leftarrow sta_j + [p_g]$;
17:             Add a new child node $\mathcal{N}'$ of $\mathcal{N}_j$ with state $sta_g$;
18:         **end for**
19:     **end if**
20:     *# Simulation*
21:     **while** $i<K$ **do**
22:         $p_i \leftarrow$ random sampling a prompt in $PS_i$;
23:         $sta_j \leftarrow sta_j + [p_i]$;
24:         update the new state $sta_j$ of $\mathcal{N}_j$;
25:     **end while**
26:     $\boldsymbol{P}_j \leftarrow$ get the prompt storage of $sta_j$;
27:     $(\boldsymbol{SO}_j, fit_j) \leftarrow$ get the operator combination and score of $\mathcal{N}_j$ by OperatorRotationEvaluation($\boldsymbol{P}_j$);
28:     *# Backpropagation*
29:     $\mathcal{N}_p \leftarrow \mathcal{N}_j$;
30:     **while** If the node $\mathcal{N}_p$ exists **do**
31:         $sco_j \leftarrow sco_j + fit_j$; *# Default node score is 0.*
32:         $vs_j \leftarrow vs_j + 1$; *# Default visit count is 0.*
33:         $\mathcal{N}_p \leftarrow$ get the parent node of $\mathcal{N}_j$;
34:     **end while**
35: **end for**
36: $\boldsymbol{O}_{best} \leftarrow$ get the highest score combination in $\boldsymbol{SO}$;
37: $\boldsymbol{P}_{best} \leftarrow$ get the state of the highest scoring node;

---

each iteration, the design prompt (containing design thoughts) for the corresponding operator is extracted from the design strategy, and the algorithm generator produces candidate algorithms. Notably, the algorithm generator can incorporate state-of-the-art AHD modules, implying that parameters must be customized for different algorithm design tasks. The generated operators are then evaluated by a multi-objective evaluator, where the average HV performance obtained over multiple runs of NSGA-II with the integrated operators serves as the fitness score. If a superior operator is found, it replaces the original operator in the combination, and the process continues to optimize the next operator. Through repeated iterations, multiple operators are rapidly rotated and evaluated, thereby obtaining scores for the design strategy. These scores are used to update the branching information between nodes in the search tree.

### D.4. Prompt Engineering

The prompts used in E2OC are for algorithm and prompt generation. During algorithm generation, E2OC adopts heuristic prompting strategies consistent with those used in existing single-heuristic design methods. For example, EoH employs

**Algorithm 3** Operator Rotation Evolution Mechanism.
___
1: **Input:** Max number of middle iterations $N_{middle}$; Number of operators to be evolved $K$; Operator design prompt tuple $\boldsymbol{P}$; Instance set $D$; Initial operator combination $\boldsymbol{O}_1$.
2: **Output:** The $(\boldsymbol{BO}, fit)$ of the highest fitness.
3: Initialize evaluator of multi-objective optimization $Eva$;
4: $fit \leftarrow Eva(D|\boldsymbol{O}_1)$;
5: $\boldsymbol{BO} \leftarrow \boldsymbol{O}_1$;
6: **for** $k = 1, \ldots, N_{middle}$ **do**
7:     **for** $i = 1, \ldots, K$ **do**
8:         $p_i \leftarrow$ get the prompt of operator $i$ in $\boldsymbol{P}$;
9:         $OS_i \leftarrow$ generate operators set by $\mathcal{G}_i(\cdot \mid p_i)$;
10:         $O'_i \leftarrow$ get the highest fitness operator in $OS_i$;
11:         $\boldsymbol{O}' \leftarrow$ update $i$-operator with $O'_i$;
12:         $fit' \leftarrow Eva(D|\boldsymbol{O}')$;
13:         **if** $fit' \geq fit$ **then**
14:             $fit \leftarrow fit'$;
15:             $\boldsymbol{BO} \leftarrow \boldsymbol{O}'$;
16:         **end if**
17:     **end for**
18: **end for**
___

evolution prompts including Exploration prompts (E1, E2) and Modification prompts (M1, M2, M3) (Liu et al., 2024a), while MCTS-AHD uses prompts such as i1, e1, e2, m1, m2, and s1 for MCTS initialization and tree expansion (Zheng et al., 2025a). Other baseline methods similarly follow the prompt strategies specified in their original studies (Romera-Paredes et al., 2024; Yao et al., 2025; Ye et al., 2024).

When constructing the language space of design strategies, dedicated prompts are used to analyze existing design thoughts and reformulate them, as illustrated in Figure 5. To ensure compatibility across different algorithm design frameworks, design strategies are embedded into structured algorithm-parameter description blocks, corresponding to the improvement suggestions shown in the figure. These prompts typically include three components. First, **Elite Candidate Operator** (`new_alg`) obtained from the warm-start phase are provided as reference targets. Second, an **Expert-designed Operator** (`ini_template`) designed by domain experts is supplied to ground the design process. Third, an Output Femplate (`output_template`) is specified to standardize the format of the LLM responses. As highlighted in the improvement suggestion task description in Figure 5, the LLM is guided to analyze the strengths and weaknesses of the reference algorithm. The model then produces a revised algorithm template augmented with explicit improvement suggestions. Each such template represents a distinct design strategy and is subsequently incorporated into the prompt strategies of different algorithm design frameworks for downstream use.

## E. Different MCTS variants

From the perspective of multi-variable optimization, the application of MCTS to multi-operator co-design can be categorized into four types based on state representation and expansion mechanisms, as illustrated in Figure 6.

- **Progressive multi-operator search**, where each node represents a single operator. The final operator set is obtained by selecting the highest-scoring path whose depth equals the total number of operators. During expansion, an algorithm generator is invoked to generate new operators as child nodes.

- **Progressive design strategy search with tuple states**, where each node represents a tuple of design rationales across operators. Expansion is performed by randomly modifying one element of the tuple, yielding a new candidate strategy. The highest-scoring tuple is then used for multi-operator design.

- **Progressive sampling and search of design thoughts**, where each node corresponds to a design thought for a single operator. During expansion, the LLM is queried to generate new thoughts for the target operator, and the highest-scoring root-to-leaf path defines the final design strategy.

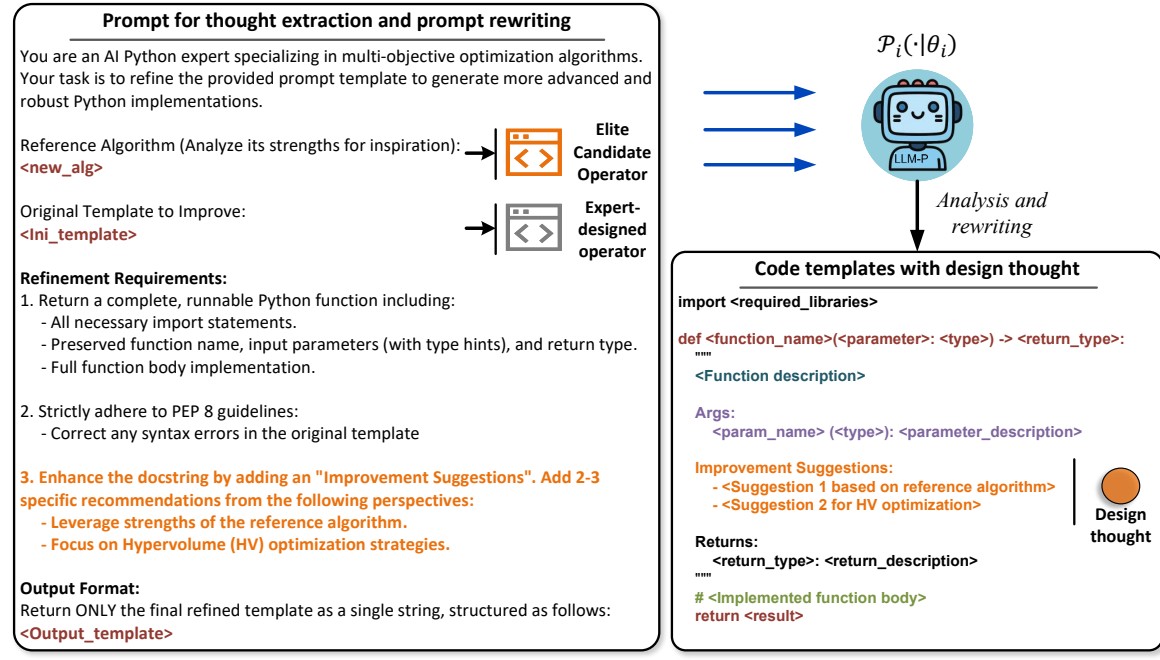

*Figure 5.* Prompt for design thought analysis and prompt rewriting.

- **Progressive design strategy search with warm-start**, corresponding to E2OC, where nodes represent individual design thoughts but the thought space at each depth is predefined during the warm-start phase and does not grow dynamically during expansion.

All four variants are capable of supporting collaborative multi-operator design. However, for thought-based search strategies (b–d in Figure 6), an additional operator design stage is required once a complete design strategy has been identified. A comparison and analysis of these variants can be found in Seciton 5.1.

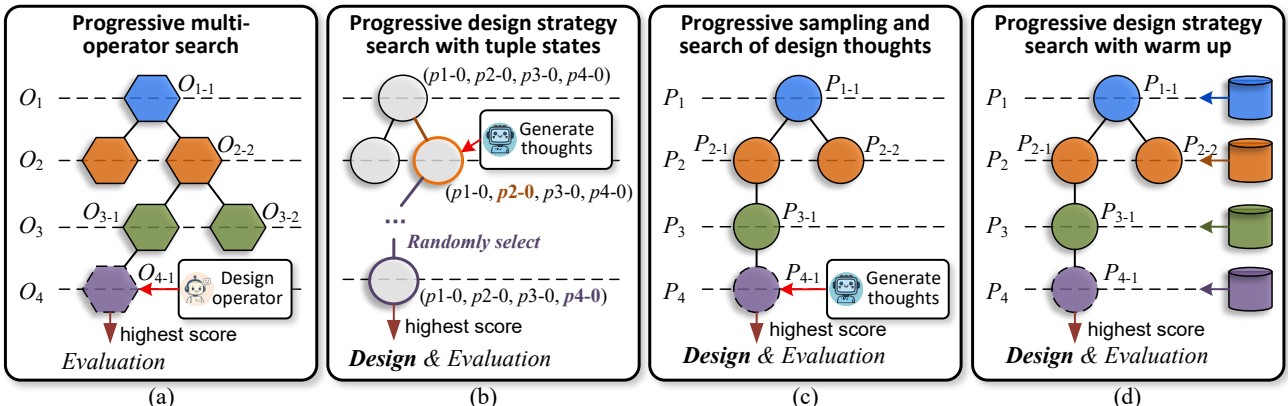

*Figure 6.* Different MCTS variants for multi-operator design.

# F. Multi-Objective Flexible Job Shop Scheduling Problem

## F.1. Problem Description

The FJSP extends the classical job shop scheduling problem by allowing each operation to be processed on multiple eligible machines with varying processing times. In multi-objective FJSPs, the objectives typically include minimizing the makespan, the maximum machine load, and the total machine load, which reflect distinct and often conflicting performance criteria.

While makespan measures the completion time of the last job, maximum machine load emphasizes the balance of heavily loaded machines, and total load captures overall resource utilization. In the Bi-FJSP, we focus on makespan and maximum machine load, whereas the Tri-FJSP additionally considers total machine load. The conflicting nature of these objectives complicates scheduling, as improvements in one criterion may degrade others.

To tackle these challenges, MOEAs are commonly employed, leveraging sophisticated operator designs for both operation sequencing and machine assignment. Designing effective operators is particularly demanding due to the combinatorial complexity, the interdependence between operations and machines, and the need to balance exploration and exploitation across objectives. The benchmark instances proposed by Brandimarte provide a standard testing platform for evaluating algorithm performance (Brandimarte, 1993). The combination of multiple objectives, practical constraints, and operator design complexity makes two- and three-objective FJSP a highly challenging setting for advanced multi-objective optimization methods.

### F.2. Operator Implementation Details

The MOEAs used to solve the multi-objective FJSP employ four operators to explore the solution space within the two-part encoding, as shown in Figure 7. Two operators target operation sequencing, performing crossover and mutation on the first part of the encoding to optimize the order of operations across jobs. The other two operators focus on machine assignment, applying crossover and mutation to the second part to refine machine selection for each operation. Each operator addresses specific optimization tasks within its respective encoding segment and is designed according to its functional requirements.

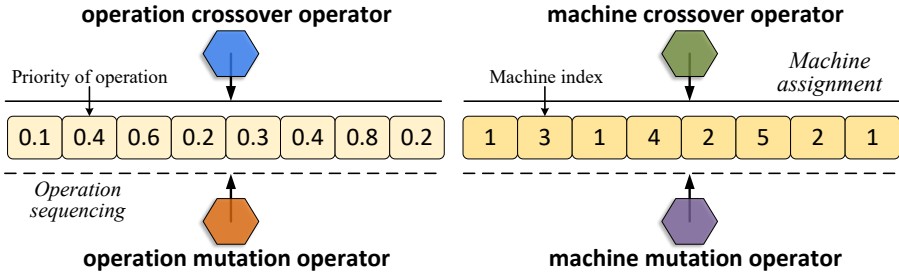

*Figure 7.* Encoding representation and operators in FJSP.

These operators exhibit interdependencies, as modifications in operation sequencing affect the performance of machine assignment, and vice versa. Designing these operators independently often leads to suboptimal performance, since improvements achieved by one operator may be offset or invalidated by another. The complex couplings among operators make it difficult to achieve balanced exploration and exploitation across multiple objectives while maintaining feasibility. Consequently, effective multi-operator evolutionary algorithm design requires mechanisms that consider operator interactions and enable their coordinated evolution to ensure consistent and robust performance in both two- and three-objective FJSP scenarios.

## G. Multi-Objective Traveling Salesman Problem

### G.1. Problem Description

The Traveling Salesman Problem (TSP) is a classical combinatorial optimization problem, where a salesman must visit a set of cities exactly once and return to the starting point, minimizing the total travel distance. In multi-objective TSPs, the objectives are typically derived by applying different weights to the distance, allowing the formulation of two- or three-objective problems that reflect trade-offs among alternative optimization criteria. While the specific objectives may vary, they are inherently conflicting, as improvements along one weighted distance can lead to deteriorations in others. Benchmark instances from publicly available datasets, such as TSPLIB, are commonly used to evaluate the performance of MOEAs in this context.

### G.2. Operator Implementation Details

To explore the solution space of multi-objective TSPs, MOEAs employ multiple domain-specific search operators that act on the same path representation, as shown in Figure 8. These operators, including crossover and mutation variants, have

**Neighborhood search operator combination**

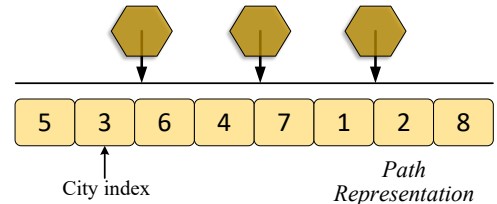

*Figure 8.* Encoding representation and operators in TSP.

overlapping functionalities but differ in the manner and scope of exploration. Each operator is designed to improve certain aspects of the tour, such as segment reordering, edge exchange, or route inversion. Despite acting on the same encoding region, the operators are interdependent: the effect of one operator may enhance or interfere with the effect of another. This overlapping and mutually influencing behavior makes independent design of operators insufficient and may result in suboptimal performance if interactions are ignored. Effectively coordinating these operators to balance exploration and exploitation across multiple objectives remains a significant challenge in multi-operator evolutionary algorithm design.

## H. Metric Definition

To evaluate the effectiveness of the proposed MOEAs, this study primarily employs two widely accepted performance metrics, hypervolume (HV) and inverted generational distance (IGD). Higher HV values and lower IGD values indicate better overall convergence and diversity performance. These metrics jointly assess the convergence and diversity of the obtained Pareto fronts, providing a comprehensive measure of algorithm performance in multi-objective optimization tasks. The following sections provide detailed definitions and formulations of HV and IGD.

### H.1. HV

Hypervolume is a widely used performance metric in multi-objective optimization that measures the volume of the objective space dominated by the obtained Pareto front relative to a reference point. A larger hypervolume indicates better convergence and diversity of solutions. In this study, the reference point $r$ is manually set to ensure it dominates all obtained solutions. Formally, given a set of Pareto-optimal solutions $P = \{p_1, p_2, \ldots, p_n\}$, the hypervolume is defined as

$$\mathrm{HV}(P) = \mathrm{vol}\Big( \bigcup_{p \in P} [p, r] \Big), \tag{10}$$

where $[p, r]$ denotes the hyper-rectangle spanned by solution $p$ and the reference point $r$, and $\mathrm{vol}(\cdot)$ represents the Lebesgue measure in the corresponding objective space.

### H.2. IGD

Inverted generational distance evaluates both convergence and diversity of an obtained Pareto front by measuring its average distance to a reference Pareto front. In this study, the reference Pareto front $P^* = \{p_1^*, p_2^*, \ldots, p_m^*\}$ is constructed from the union of non-dominated solutions obtained by all compared algorithms to approximate the true Pareto front. Given an obtained solution set $P = \{p_1, p_2, \ldots, p_n\}$, the IGD is computed as

$$\mathrm{IGD}(P, P^*) = \frac{1}{|P^*|} \sum_{p^* \in P^*} \min_{p \in P} d(p, p^*), \tag{11}$$

where $d(p, p^*)$ is the Euclidean distance between solution $p$ and reference solution $p^*$ in the objective space. Lower IGD values indicate that the obtained solutions are closer to and more uniformly distributed along the reference Pareto front.

## H.3. RI

The Relative Improvement (RI) metric quantifies the percentage improvement of a new method's performance relative to a baseline method. It is calculated using the formula:

$$RI = \frac{A - B}{B} \times 100\% \tag{12}$$

where $A$ represents the performance value of the new method and $B$ represents the performance value of the baseline method.

# I. Experiment Design and Implementation Detail

## I.1. Experimental Design.

The proposed E2OC is used to co-design multi-operators in MOEAs, which reduces manual design effort and enhances algorithmic performance. The key research questions are as follows: (1) Can automatically designed operator combinations outperform expert-designed counterparts? (2) Can high efficiency be maintained across additional objectives and diverse problem instances? (3) Does E2OC exhibit superior performance compared to advanced methods that evolve operators independently? (4) Is E2OC more efficient than alternative multi-operator design strategies? (5) Does the approach possess potential for continuous optimization? (6) Are all constituent modules of E2OC effective?

To rigorously address these questions, the following experiments are designed: (1) comparison with state-of-the-art multi-objective evolutionary algorithms on bi-objective and tri-objective FJSP and TSP; (2) comparison with recent single-heuristic automatic design methods; (3) evaluation against different multi-operator design strategies; (4) evaluate the potential for continuous optimization of E2OC across multiple iterations; (5) analysis of performance and computational cost under varying LLMs and key parameter $AP$ (Number of initial added prompt); (6) ablation studies. Moreover, in order to observe the improvement brought by the designed operator combinations over the classical operators, We compare them on several MOEAs and analyze the effect of different combinations and orders of classical operators on optimization performance.

## I.2. Comparison Method Detail

In addition to recent SOTA single-heuristic design methods, this study compares E2OC with a range of multi-heuristic design frameworks. These methods can be categorized along three dimensions, namely whether algorithmic thoughts are explicitly incorporated, whether prompt rewriting is employed, and whether a warm-start phase is required, as summarized in Table 6. Single-heuristic methods focus on designing a single algorithm and therefore do not rely on warm-start mechanisms to balance exploration and exploitation across multiple design tasks.

*Table 6.* Comparison of different methods on ideas inclusion, prompt rewriting, and warm-start

| Type | Methods | Thoughts | Prompts | Warm-start |
|---|---|---|---|---|
| Single-heuristic | Random | ✗ | ✗ | ✗ |
| | FunSearch | ✗ | ✗ | ✗ |
| | EoH | ✓ | ✗ | ✗ |
| | MEoH | ✓ | ✗ | ✗ |
| | ReEvo | ✗ | ✓ | ✗ |
| | MCTS-AHD | ✓ | ✗ | ✗ |
| Multi-heuristic | CD | ✓ | ✗ | ✗ |
| | UCB | ✓ | ✗ | ✓ |
| | Win-UCB | ✓ | ✓ | ✓ |
| | LLM | ✓ | ✓ | ✗ |
| | MCTS_OC | ✓ | ✗ | ✗ |
| | MCTS_Tuple | ✓ | ✓ | ✓ |
| | MCTS_Sampling | ✓ | ✓ | ✓ |
| | E2OC | ✓ | ✓ | ✓ |

In this context, an *idea* is defined as a language description that represents the high-level logic of a heuristic (Liu et al., 2024a). Methods such as EoH (Liu et al., 2024a), MEOH (Yao et al., 2025), and MCTS-AHD (Zheng et al., 2025a) manage ideas together with executable code as part of the population archive. Among the compared approaches, only ReEvo explicitly incorporates prompt rewriting. Empirical evidence suggests that dynamic prompt adjustment is effective in reducing code generation errors and improving the quality of generated algorithms.

Existing LLM-based collaborative algorithm design frameworks can be categorized according to how they formulate the multi-heuristic design task and organize decision-making over interacting algorithms. From this perspective, prior studies can be broadly divided into four classes: coordinate-descent (CD)-based (Wright, 2015), upper-confidence-bound(UCB)-based (Garivier & Cappé, 2011; Gupta et al., 2021), MCTS-based (Zheng et al., 2025a; Kiet et al., 2025), and LLM-driven approaches.

Coordinate-descent-based methods formulate multi-heuristic collaboration as a deterministic continuous optimization problem over the algorithm space (Wright, 2015). Multiple algorithms are optimized in a rotational manner, where one algorithm is updated while others are fixed, and the process iterates to identify high-performing combinations, corresponding to **CD** in Table 6. These methods emphasize structured and controllable search dynamics and are most effective when inter-heuristic interactions are relatively stable.

UCB-based methods model collaborative algorithm design as a stochastic decision-making problem (Garivier & Cappé, 2011), typically using a multi-armed bandit formulation. Each algorithm or operator is treated as an arm with an unknown reward distribution, and the search explicitly balances exploration and exploitation **UCB**, corresponding to in Table 6. When LLM-generated prompts are introduced, reward distributions may shift over time, complicating estimation. Moreover, dependencies among algorithms often motivate extensions to combinatorial bandit settings (Gupta et al., 2021), where window-constrained UCB strategies are used to adapt to non-stationary environments, corresponding to **Win-UCB** in Table 6.

MCTS-based methods cast multi-heuristic design as a sequential decision-making process, in which each design action influences subsequent states. A search tree is incrementally constructed to encode historical design trajectories, enabling a principled trade-off between exploration and exploitation. To control search complexity and handle non-stationary feedback, practical implementations commonly restrict search horizons or limit tree depth. This paradigm has been applied to both single-heuristic design and collaborative search over algorithm combinations or design strategies, corresponding to **MCTS-OC, MCTS-Tuple and MCTS-Sampling** in Table 6.

Finally, fully LLM-driven approaches dispense with explicit search heuristics and rely on structured prompts to enable LLMs to autonomously perform operator selection, algorithm composition, and resource allocation. These methods exploit high-level semantic reasoning to dynamically adjust collaborative optimization strategies, offering greater flexibility at the cost of reduced explicit control, corresponding to **LLM** in Table 6.

Notably, most multi-heuristic design frameworks treat single-heuristic design methods as modular building blocks. In this study, EoH is adopted as the foundational single-heuristic design module across all compared frameworks, with explicit management of algorithm thoughts. Among these frameworks, UCB-based methods and those constructing explicit design spaces typically require a warm-start phase for initialization.

### I.3. Other Parameter Settings

In the offline design phase, the deepseek-chat model is selected based on quality-cost performance, and all model temperature values default to 1. The proposed E2OC can be divided into four components: the outer MCTS, the middle operator rotation and the inner algorithm generator and evaluator, with hyperparameter settings specifically shown in Table 7. The offline evaluators are used to rapidly assess newly designed operators and are configured with half of the computational budget used in online evaluations.

Specifically, for the FJSP, the offline setting uses 15 iterations and a population size of 50, while for the TSP it uses 100 iterations and a population size of 100. Both problems are evaluated three times in the offline stage to achieve a rapid assessment.

The online evaluation settings follow established practices in prior studies. For the FJSP, 30 iterations and a population size of 200 are used. For the TSP, 200 iterations and a population size of 200 are adopted, with five independent runs conducted for each configuration. The mean performance over all runs is reported as the final performance of each multi-objective optimization algorithm.

### I.4. Resource Consumption

The authors of ReEvo (Ye et al., 2024) argued that efficiency benchmarking for LLM-EPS methods should focus on the number of fitness evaluations rather than the number of LLM calls. Similarly, MCTS-AHD, as the most recent LLM-based

*Table 7.* Overview of hyperparameters used in E2OC on Bi-FJSP. The values of the parameters with * are defined by the experiment, all others are default values.

| Type | Component | Hyperparameters | Value |
|---|---|---|---|
| **Offline** | LLM | Model | **deepseek-chat*** |
| | | Temperature | 1.0 |
| | MCTS | Outer iteration | 30 |
| | | Number of initial operator | 4 |
| | | Number of initial added prompt | **3*** |
| | Operator Rotation | Middle iteration | 5 |
| | Generator | Inner iteration | 10 |
| | | Operator population size | 10 |
| | | Max sampling number | 25 |
| | Evaluator | Iteration | FJSP-15, TSP-30 |
| | | Solution population size | FJSP-50, TSP-100 |
| | | Number of validations | 3 |
| **Online** | Evaluation | Iteration | FJSP-30, BiTSP-200,TriTSP-100 |
| | | Solution population size | FJSP-100, TSP-200 |
| | | Number of validations | 5 |

AHD method at the time of its submission, also follows this benchmarking protocol (Zheng et al., 2025a). Accordingly, in this study, the performance of different methods is compared by controlling for a similar number of fitness evaluations, ensuring consistency in assessment.

The key algorithmic parameters of E2OC for solving FJSP are summarized in Table 7, including the outer-layer MCTS configuration, the number of operator-rotation iterations, as well as parameters related to the algorithm generator and the evaluator. Each newly designed operator combination is repeatedly embedded into MOEAs for optimization, and its performance is assessed by averaging the resulting HV or IGD values. The overall multi-heuristic design process of E2OC is realized through iterative interactions between the algorithm generator and the prompt generator.

The number of evaluations required for algorithm design equals the number of generated algorithms and is given by

$$(iter_{\text{out}} + 1) \times iter_{\text{mid}} \times K \times sam_{\text{max}}. \tag{13}$$

Here, $iter_{\text{out}} + 1$ denotes the sum of the warm-start stage and the outer MCTS iterations, $iter_{\text{mid}}$ represents the number of operator-rotation steps, and $sam_{\text{max}}$ is the maximum number of newly generated algorithms accumulated by the internal algorithm generator. The algorithm generator supports different design modules, such as EoH and ReEvo. To eliminate the influence of heterogeneous population selection mechanisms across different generators, $sam_{\text{max}}$ is used as a unified upper bound on the number of algorithms generated per design task, while the remaining parameters follow the settings reported in the corresponding literature.

Compared with other methods, E2OC additionally relies on the prompt generator to construct the design strategy space for operator combinations, which incurs $K \times AP$ calls to the LLM interface.

### I.5. MOEAs Parameter Settings

Directly applying newly designed operator combinations in MOEAs to optimize MCOPs does not yield reliable quantitative performance. Instead, as shown in Table 7 regarding the number of verifications, multiple validations are required, and performance must be assessed based on the aggregated results of these repeated evaluations, which incurs higher computational costs.

In this study, three classical multi-objective evolutionary algorithms (MOEAs) are employed as baseline methods: NSGA-II (Deb et al., 2002), NSGA-III (Deb & Jain, 2014), and MOEA/D (Qingfu Zhang & Hui Li, 2007). The key parameter settings are summarized below, serving as default values for all benchmark experiments. These settings can be adjusted according to problem scale and complexity.

- **NSGA-II and NSGA-III:** For Bi-FJSP and Tri-FJSP, the population size is set to 100 with a maximum of 250 generations. For Bi-TSP and Tri-TSP, the population size and maximum generations are set to 100 and 250, respectively.

- **MOEA/D:** The population size is set to 150, with a maximum of 200 generations. The neighborhood size is 20, and the probability of selecting individuals from the neighborhood is 0.9.

All algorithms employ the same initial neighborhood operators. For FJSP, Simulated Binary Crossover (SBX) and polynomial

mutation are used with consistent crossover and mutation probabilities. For TSP, the local search operators OR-Opt, 2-Opt, and 3-Opt are applied. Reference points are set identically across all benchmark instances. These parameter settings ensure a reasonable balance between exploration and exploitation across all MOEAs while maintaining consistency for fair comparisons in benchmark evaluations.

### I.6. Implementation of different AHD methods

To ensure a fair and consistent comparison with existing LLM-based automated algorithm design methods, we normalize the computational budget across all competing approaches using a unified algorithm evaluation resources.

**Single-heuristic Design Methods.** When comparing against single-heuristic design methods, the multi-heuristic design problem is decomposed into a sequence of independent single-heuristic design tasks. The total evaluation budget is fixed and evenly distributed across these sub-tasks. For EoH, the population size is set to 20, consistent with the original implementation, while the algorithm terminates upon reaching a predefined maximum number of sampled candidates rather than a fixed number of generations. ReEvo explicitly constrains the number of newly constructed prompts. To ensure comparability, this limit is set to $K \times AP$, matching the prompt budget used in E2OC. All remaining baseline methods adopt the parameter settings recommended in their respective studies and are likewise terminated based on the maximum number of sampled designs.

**Multi-heuristic Design Methods.** For the comparison with multi-heuristic design frameworks, we still use the same total evaluation budget. To accurately compare the performance of different multi-heuristic search frameworks, we ensure that the evaluation resources for each operator design task are consistent and that all evaluations are performed on EoH. Specifically, the maximum number of evaluations allocated to a single algorithm design task within one decision round is fixed and defined as a *standard design resource*. This definition enables a principled comparison across frameworks with fundamentally different control structures.

In CD framework, the number of rotation iterations determines how many times algorithms are optimized in an alternating manner. Within each rotation, operators are designed sequentially, and the design of one operator consumes exactly one standard design resource. Accordingly, the total number of rotation steps is set to $(iter_{out} + 1) \times iter_{mid}$, which aligns the overall resource usage with that of E2OC.

UCB-based framework do not follow a predetermined design order but instead dynamically select algorithms based on estimated utility. Under the unified resource definition, the total number of available standard design resource is set to $(iter_{out} + 1) \times iter_{mid} \times K$, reflecting the additional flexibility introduced by operator-level selection.

Among the MCTS-based variants, MCTS_OC does not perform explicit search over design strategy spaces. As a result, its effective outer-loop iteration count is set to $(iter_{out} + 1) \times iter_{mid} \times K$, where each tree node corresponds to one standard design resource. In contrast, MCTS-Tuple and MCTS-Fixing explicitly explore strategy-level decision spaces and therefore adopt the same parameter settings as E2OC.

It is worth noting that the parameter settings of the LLM-based AHD methods are consistent with those of UCB-related methods. Under these unified resource allocation rules, all methods are evaluated with an equivalent number of standard design resources, ensuring that observed performance differences can be attributed to the quality of the multi-heuristic search mechanisms rather than disparities in evaluation budgets.

## J. Additional Experiment Results

### J.1. More Visualization Results

This section supplements the visualization results of the experiments. Comparisons with the convergence processes and PF of classical operators in different MOEAs, different AHD methods, and different MCTS variants methods in training and testing instances are included.

**Comparison with Expert Design Operators.** The experimental performance of the multi-operator designed by E2OC compared with classic operator combinations across different MOEAs is shown in Table 1. E2OC builds upon existing expert operators (serving as the initial operator combination) to further customize and design superior operator combinations for different methods. The HV, IGD convergence process, and PF for different methods on Bi-FJSP and Tri-TSP in this

experiment are illustrated in Figure 9 and 10.

In the Bi-FJSP experiments, it can be observed that, except for the training instance mk15, the operators designed by E2OC continue to perform remarkably well in the other two similar instances in Figure 9. In terms of the PF, the operator combinations designed by E2OC achieve greater diversity, which fully demonstrates their superior performance and generalization capability.

In the Tri-TSP experiments, all methods achieve relatively similar performance within 100 iterations on the 100-node training set. However, it is also evident from the PF that the operator combinations designed by E2OC (represented by the blue-green point set) exhibit significantly greater diversity and occupy a superior region in Figure 10. The improvement is particularly notable in small-scale scenarios, where both the HV convergence curves and PF distributions clearly outperform the initial operators designed by human experts.

**Comparison with AHD Methods.** Experimental results are summarized in Table 6, while Fig. 11 illustrates the convergence trends of HV and IGD, as well as the final PF obtained by different AHD methods on both training and testing instances. It is worth noting that the evaluation phase is conducted on both the training set and the test set, where a larger population size and more iterations are used than in the design phase. In Bi-FJSP, the testing instance is selected to evaluate the algorithms on mk13 or mk14, which have larger number of processes and devices. All operator designs are executed 5 independent runs in NSGA-II, and performance is reported as the average over these runs. The PF shown in the figure corresponds to the run that finds the most non-dominated solutions out of the five repeated runs.

The results reveal that LLM-based multi-operator co-search methods achieve relatively good performance on training data, but their performance degrades sharply on testing instances. It converts decision information, such as the historical performance characteristics of different operators, into language descriptions and rely directly on the LLM to make operator-search decisions. Although prolonged iteration can lead to strong training-set performance, such approaches struggle to maintain effectiveness in new scenarios.

In addition to a multi-operator codesign framework that allows dynamic modification of prompts, such as Win-UCB and E2OC, classical AHD methods yield similar outcomes on both training and testing instances under the same evaluation budget, which indicates their tendency to converge toward local optima in algorithm design. By adjusting its prompts, Win-UCB achieves the second-best HV performance on the training set. This result shows that modifying prompts and design thoughts during the search process can help escape local optima and lead to better-performing algorithms.

E2OC achieves the best performance on both training and testing instances, confirming that constructing a prior design-knowledge surface and performing local search at the higher strategic domain is more effective than operating under a fixed algorithmic generation distribution(fixed prompts or code templates). Furthermore, the optimization performance of E2OC on new instances is significantly better than that of other methods.

**Comparison with Different Methods of MCTS variants.** The experimental results of different MCTS variants are summarized in Table 5. Fig. 12 further illustrates the convergence trends of HV and IGD as well as the final Pareto front (PF) obtained by these methods on both training and testing instances. The plotting settings follow those described in Section J.1.

The results show that MCTS_OC, which directly searches operator combinations, performs poorly. This suggests that under fixed prompts or code templates, the algorithm is prone to converge to local optima, with limited ability to break through the existing generation distribution. MCTS_Tuple and MCTS_Sample differ in their node-state representations, but both allow unrestricted dynamic sampling of new design thoughts. While MCTS_Tuple achieves better performance on the training set, the unrestricted sampling continuously updates branch information in the Monte Carlo tree, making it difficult to identify superior combinations within a limited evaluation budget. In contrast, E2OC consistently delivers superior performance, indicating that progressive search over a warm-start-constructed local design space is more effective.

### J.2. Different Parameters

The number of initial added prompts $AP$ controls the number of operator design thoughts generated during the warm-start phase. Larger $AP$ values increase the diversity of initial design thoughts and cover a broader range of improvement directions, but they also expand the design strategy search space, making it more difficult to identify optimal strategy paths.

With four operators, each associated with $AP$ design thoughts, the number of possible design strategies scales as $AP^4$. All experiments use the same manually designed initial operator code templates, design thoughts, and warm-start operator

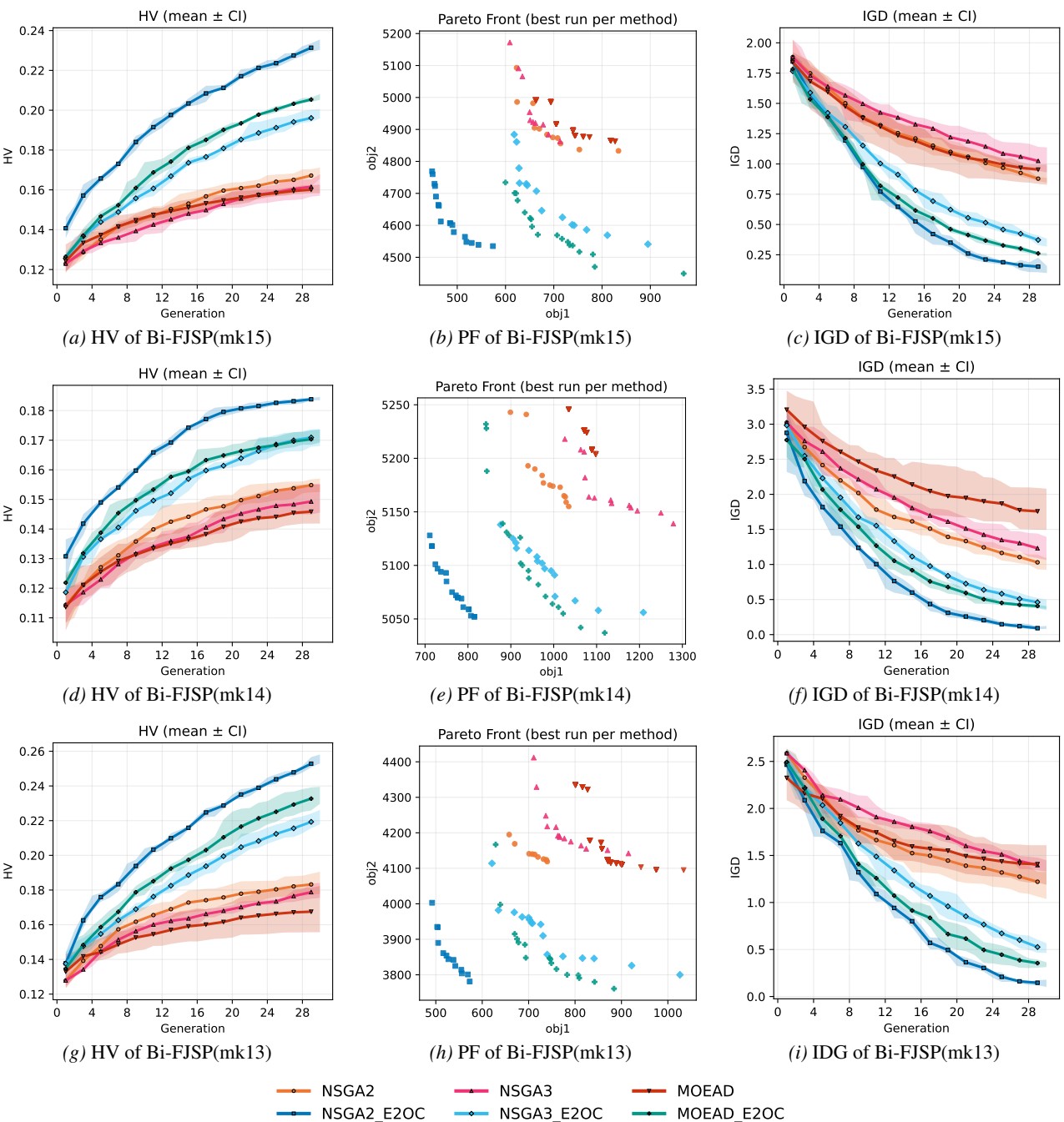

*Figure 9.* HV, IDG and PF performance of different operators in MOEAs on Bi-FJSP.

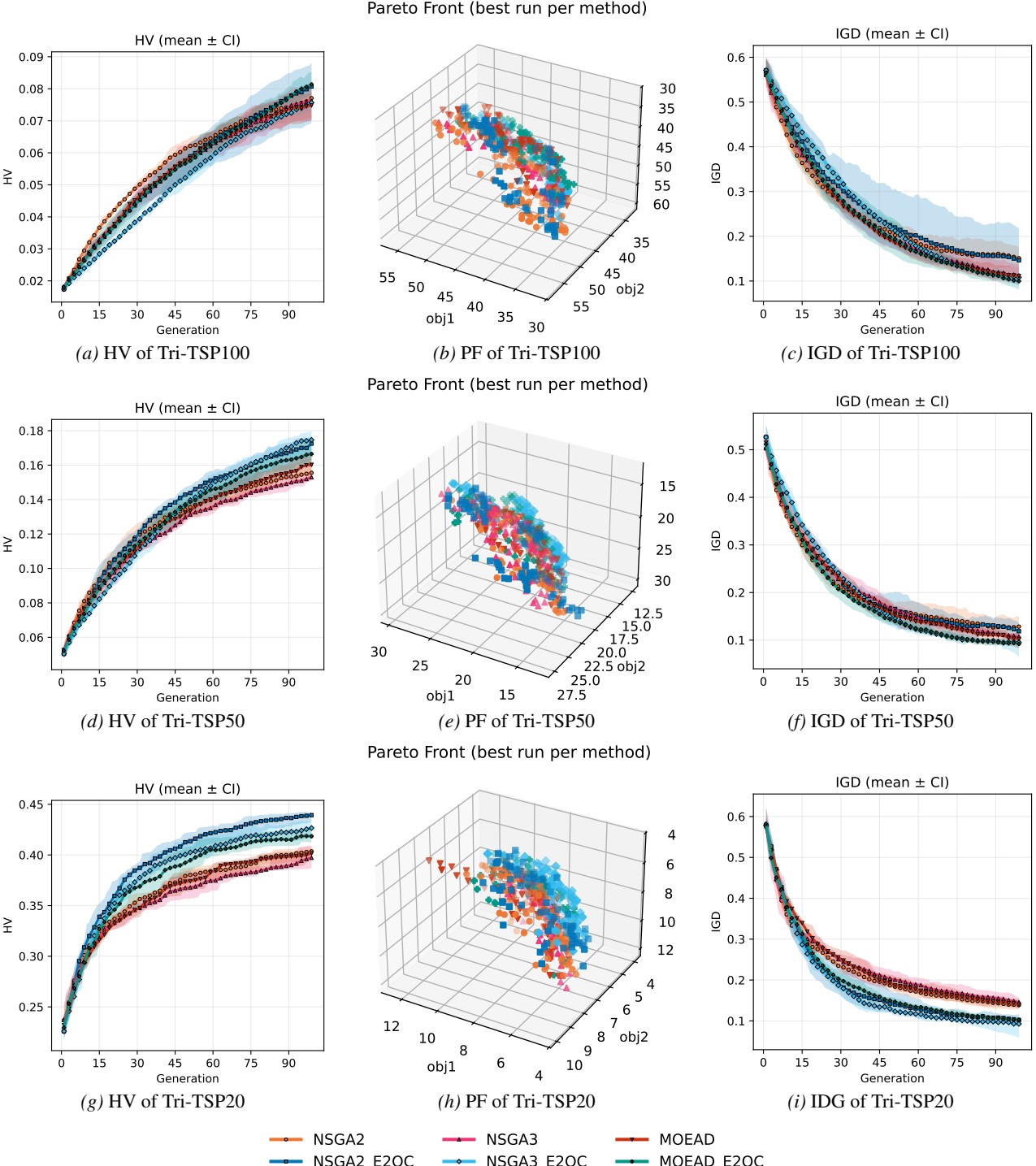

*Figure 10.* HV, IDG and PF performance of different operators in MOEAs on Tri-TSP.

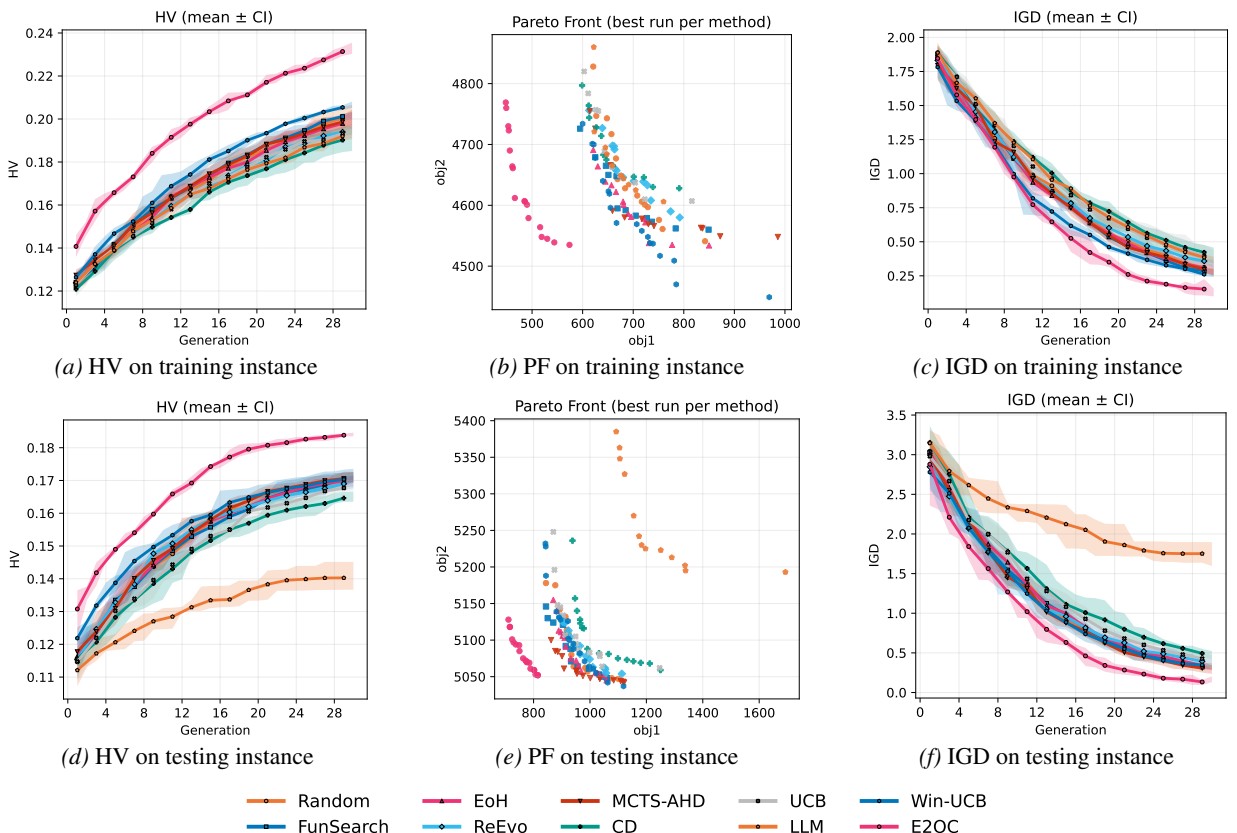

*Figure 11.* Performance of different AHD methods on BIFJSP training (mk15) and testing (mk14) instances.

sets, and $AP$ is varied solely to control the number of LLM-generated design thoughts. We evaluate $AP \in [1, 3, 5, 7]$, corresponding to strategy spaces of sizes $[4^2, 4^4, 4^6, 4^8]$, respectively.

To assess the impact of different $AP$ settings, instances are split into training and testing sets, and the search behavior and performance of generated operators are analyzed. The results, summarized in Table 8, show that larger $AP$ values increase the difficulty of invalid branches caused by the pruning conflict design strategy in MCTS, leading to a higher proportion of illegal operator code. Although $AP = 5$ achieves the best HV on the training set, $AP = 3$ delivers the most stable and best overall performance across all instances and the test set. To avoid overfitting, $AP = 3$ is adopted in all subsequent experiments.

*Table 8.* Comparison of different number of initial added prompt $AP$ (see Algorithm 1). The performance metrics of the search process include: ValidRate (correctness of generated code), Mean, Range, and standard values Std of the scores.

| Parameter | $AP$ | 1 | 3 | 5 | 7 |
|---|---|---|---|---|---|
| **Search** | ValidR↑ | **0.9969** | 0.9965 | 0.9946 | 0.9930 |
| **performance** | Mean↑ | 0.1508 | **0.1574** | 0.1498 | 0.1511 |
| | Range↑ | 0.1638 | **0.1749** | 0.1639 | 0.1654 |
| | Std↑ | 0.0087 | 0.0120 | 0.0098 | **0.0127** |
| **All instances** | HV↑ | 0.2199 | **0.2260** | 0.2198 | 0.2185 |
| | IGD↓ | 1.5762 | **1.3966** | 1.5782 | 1.5891 |
| **Train instance** | HV↑ | 0.1732 | 0.1709 | **0.1746** | 0.1726 |
| | IGD↓ | 0.7796 | 0.8532 | **0.7734** | 0.7735 |
| **Test instances** | HV↑ | 0.2232 | **0.2300** | 0.2230 | 0.2218 |
| | IGD↓ | 1.6331 | **1.4355** | 1.6357 | 1.6474 |

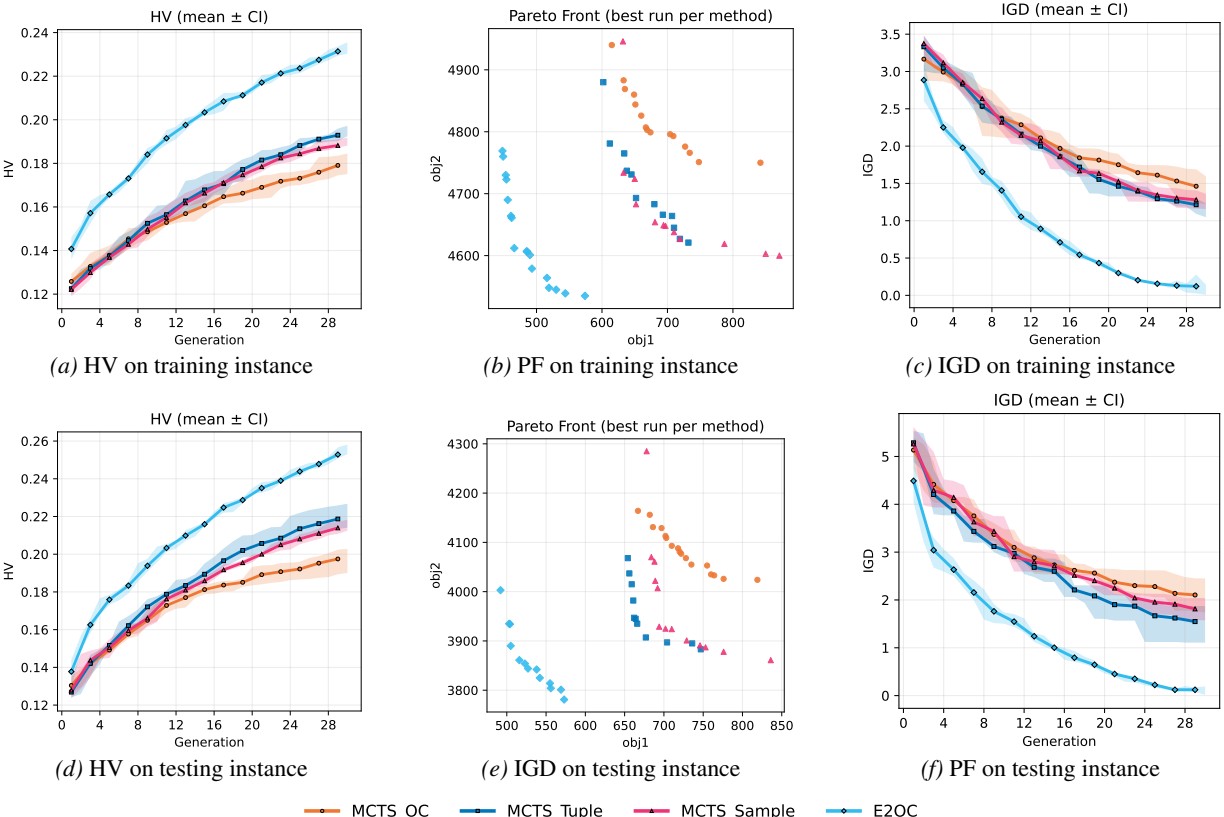

*Figure 12.* Performance of different MCTS variants on BIFJSP training (mk15) and testing (mk13) instances.

## J.3. Comparison with Expert-Designed Operators

To systematically assess the performance gap between E2OC and expert-designed operators, we conduct comparative experiments against classical operators and their manually constructed combinations on the TSP. NSGA-II is adopted as the multi-objective baseline algorithm. The evaluated operator set includes classical local search heuristics for TSP, named **2opt**, **3opt**, and **oropt**, as well as commonly used genetic operators such as ox and swap. These operators are organized into three categories. The first category consists of individual operators (2opt, 3opt, and oropt). The second category includes hybrid combinations of crossover, mutation, and local search, namely **ox_swap**, **ox_swap_2opt**, **ox_swap_3opt**, and **ox_swap_oropt**. The third category contains sequential compositions of classical local search operators with different execution orders, including **2opt_3opt_oropt**, **3opt_2opt_oropt**, and **oropt_2opt_3opt**. Among these, the combination oropt_2opt_3opt exhibits the worst average performance and is therefore selected as the initial operator configuration for E2OC. The RI is measured with respect to this baseline, following the definition in Section H.3. The experimental results are summarized in Table 9.

The results indicate that the standalone 2opt operator achieves the best HV performance, surpassing multiple manually designed operator combinations. The convergence trends of HV and IGD, and the Pareto front for all operators in Bi-TSP20, 50, and 100 are shown in Figure 13. Although the combination of ox and swap yields a relative improvement of 2.76% on the bi-objective TSP20 instance, further incorporating local search operators such as 2opt leads to performance degradation. To analyze the influence of operator ordering, different permutations of 2opt, 3opt, and oropt are evaluated within the third category. All reordered combinations exhibit inferior performance, with varying degrees of degradation, suggesting the presence of implicit operator incompatibilities that are difficult to resolve through manual composition. Using the consistently worst-performing oropt_2opt_3opt as the starting configuration, E2OC achieves the best performance on TSP50 and TSP100. Through a progressive search over operator design principles and composition strategies, E2OC not only substantially improves upon the initial configuration but also outperforms the best expert-designed operator, 2opt.

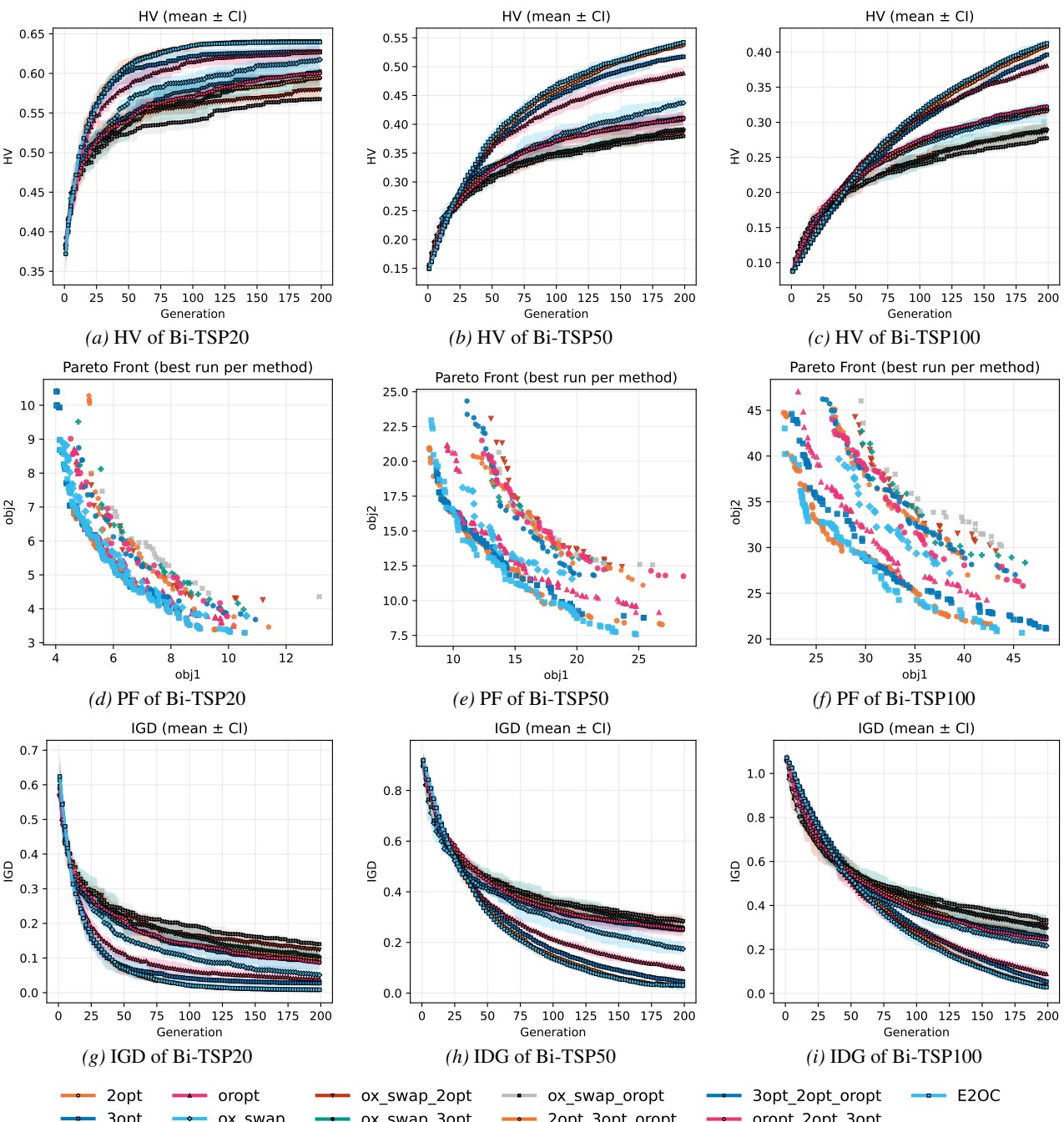

*Figure 13.* HV, IDG and PF performance of different operators in solving Bi-TSPs.

*Table 9.* Comparison of NSGA-II solving TSP with different classical operators. The ox is the sequential crossover operator and swap is the random swap mutation operator. The results are divided into three groups: group 1 is a single operator solved independently, group 2 is a cross-variable operator plus other neighborhood operators, and group 3 is a different order in the combinations. Bold text indicates optimal performance, and gray highlighting represents the baseline operator combination, which is the initial operator for E2OC.

| Method | Bi-TSP20 | | | Bi-TSP50 | | | Bi-TSP100 | | |
|---|---|---|---|---|---|---|---|---|---|
| | HV↑ | IGD↓ | RI↑ | HV↑ | IGD↓ | RI↑ | HV↑ | IGD↓ | RI↑ |
| 2opt | **0.6117±0.0523** | **0.0684±0.1124** | **9.29%** | 0.4255±0.1016 | 0.2122±0.2124 | 22.11% | 0.2882±0.0907 | 0.3322±0.2708 | 12.54% |
| 3opt | 0.6003±0.0497 | 0.0821±0.1082 | 7.26% | 0.4143±0.0971 | 0.2310±0.2074 | 18.90% | 0.2797±0.0868 | 0.3555±0.2643 | 9.22% |
| oropt | 0.5932±0.0489 | 0.1041±0.1026 | 5.99% | 0.4001±0.0839 | 0.2587±0.1850 | 14.82% | 0.2842±0.0792 | 0.3436±0.2418 | 10.98% |
| ox_swap | 0.5751±0.0473 | 0.1348±0.0994 | 2.76% | 0.3604±0.0641 | 0.3081±0.1487 | 3.43% | 0.2513±0.0554 | 0.3970±0.1823 | -1.89% |
| ox_swap_2opt | 0.5472±0.0365 | 0.1971±0.0851 | -2.22% | 0.3312±0.0510 | 0.3746±0.1219 | -4.94% | 0.2342±0.0466 | 0.4459±0.1562 | -8.55% |
| ox_swap_3opt | 0.5540±0.0420 | 0.1859±0.0870 | -1.02% | 0.3332±0.0533 | 0.3677±0.1300 | -4.36% | 0.2339±0.0481 | 0.4548±0.1638 | -8.67% |
| ox_swap_oropt | 0.5350±0.0349 | 0.2127±0.0786 | -4.41% | 0.3279±0.0496 | 0.3790±0.1217 | -5.90% | 0.2282±0.0422 | 0.4646±0.1462 | -10.89% |
| 2opt_3opt_oropt | 0.5576±0.0399 | 0.1710±0.0897 | -0.37% | 0.3505±0.0575 | 0.3618±0.1373 | 0.60% | 0.2517±0.0579 | 0.4292±0.1844 | -1.70% |
| 3opt_2opt_oropt | 0.5637±0.0428 | 0.1637±0.0894 | 0.71% | 0.3517±0.0586 | 0.3459±0.1354 | 0.94% | 0.2546±0.0581 | 0.4174±0.1847 | -0.58% |
| oropt_2opt_3opt | 0.5597±0.0430 | 0.1698±0.0929 | 0.00% | 0.3484±0.0610 | 0.3596±0.1434 | 0.00% | 0.2561±0.0584 | 0.4165±0.1871 | 0.00% |
| E2OC | 0.6104±0.0522 | 0.0710±0.1132 | 9.06% | **0.4312±0.1023** | **0.1023±0.0105** | **23.75%** | **0.2929±0.0930** | 0.3628±0.2928 | **14.38%** |

## J.4. Generalization Comparison on TSPs with Different Scales

This section further examines the generalization capability of operators designed by E2OC by comparing them with the best expert-designed operator (2opt) and the initial operator configuration (oropt_2opt_3opt) on larger-scale Bi-TSP150 and Bi-TSP200 instances, as summarized in Table 10. In Bi-TSP150 and Bi-TSP200, E2OC achieves relative improvements of approximately 30.93% and 22.06%, respectively, over oropt_2opt_3opt. In contrast, the performance of the 2opt operator deteriorates as the problem scale increases. The convergence trends of HV and IGD, and the Pareto front of different method in Bi-TSP150 and 200 are shown in Figure 14. In terms of average HV, E2OC consistently achieves the best results across small-scale instances (Bi-TSP20-200), large-scale instances (Bi-TSP150-200), and the complete benchmark set, thereby demonstrating strong generalization performance across different problem scales.

*Table 10.* Comparison of NSGA-II for solving TSP of different sizes with different operators. TSP20-100 refers to the instance set containing TSP20, 50, and 100, and the same applies to others.

| Method | Bi-TSP150 | | | Bi-TSP200 | | | Mean HV↑ | | |
|---|---|---|---|---|---|---|---|---|---|
| | HV↑ | IGD↓ | RI↑ | HV↑ | IGD↓ | RI↑ | TSP20-100 | TSP150-200 | TSP20-200 |
| 2opt | 0.1205±0.0550 | 0.5239±0.2611 | -10.48% | 0.1026±0.0442 | 0.5905±0.2845 | -18.27% | 0.4418 | 0.1116 | 0.3097 |
| oropt_2opt_3opt | 0.1346±0.0485 | 0.4477±0.2232 | 0.00% | 0.1256±0.0454 | 0.4591±0.2672 | 0.00% | 0.3881 | 0.1301 | 0.2849 |
| E2OC | **0.1762±0.0647** | 0.7430±0.5519 | **30.93%** | **0.1533±0.0554** | 0.5150±0.4603 | **22.06%** | **0.4448** | **0.1648** | **0.3328** |

## J.5. Results and Discussion on Scalability and Generalization

To validate scalability, Table 11 evaluates the performance of different frameworks on more MOPs, including Bi-CVRP and Bi-KP, across various problem dimensions using HV and IGD metrics. The parameters used in these problems, such as population size, are consistent with those in the TSP. The results indicate that E2OC consistently achieves the best HV and IGD values compared to NSGA-II and EOH across all tested instances. Crucially, this advantage becomes more pronounced

*Table 11.* Comparison on more MOPs (Bi-CVRP and Bi-KP).

| Method | Bi-CVRP20 | | | Bi-CVRP50 | | | Bi-CVRP100 | | |
|---|---|---|---|---|---|---|---|---|---|
| | HV↑ | IGD↓ | RI↑ | HV↑ | IGD↓ | RI↑ | HV↑ | IGD↓ | RI↑ |
| NSGA2 | 0.3147±0.0281 | 2.3547±0.5570 | 0.00% | 0.1488±0.0166 | 7.7848±0.8779 | 0.00% | 0.0373±0.0164 | 0.2675±0.1001 | 0.00% |
| EOH | 0.3947±0.0525 | 1.0007±0.8573 | 25.42% | 0.2200±0.0452 | 4.5495±2.0133 | 47.81% | 0.0652±0.0473 | 1.4160±0.6774 | 74.93% |
| E2OC | 0.4009±0.0567 | 0.9011±0.9396 | 27.39% | 0.2473±0.0647 | 3.4112±2.7225 | 66.16% | 0.0673±0.0515 | 1.5564±0.8211 | 80.59% |

| Method | Bi-KP50 | | | Bi-KP100 | | | Bi-KP200 | | |
|---|---|---|---|---|---|---|---|---|---|
| | HV↑ | IGD↓ | RI↑ | HV↑ | IGD↓ | RI↑ | HV↑ | IGD↓ | RI↑ |
| NSGA2 | 14.0851±0.2062 | 0.3205±0.0997 | 0.00% | 0.7650±0.0970 | 0.9104±0.3044 | 0.00% | 0.2281±0.0967 | 0.9624±0.6044 | 0.00% |
| EOH | 14.4532±0.1803 | 0.1673±0.0762 | 2.61% | 0.9792±0.1598 | 0.3763±0.4312 | 28.00% | 0.2818±0.1008 | 0.9449±0.6371 | 23.51% |
| E2OC | 14.5687±0.2108 | 0.1226±0.0915 | 3.43% | 0.9950±0.1575 | 0.3396±0.4252 | 30.06% | 0.3231±0.1410 | 0.6175±0.7478 | 41.64% |

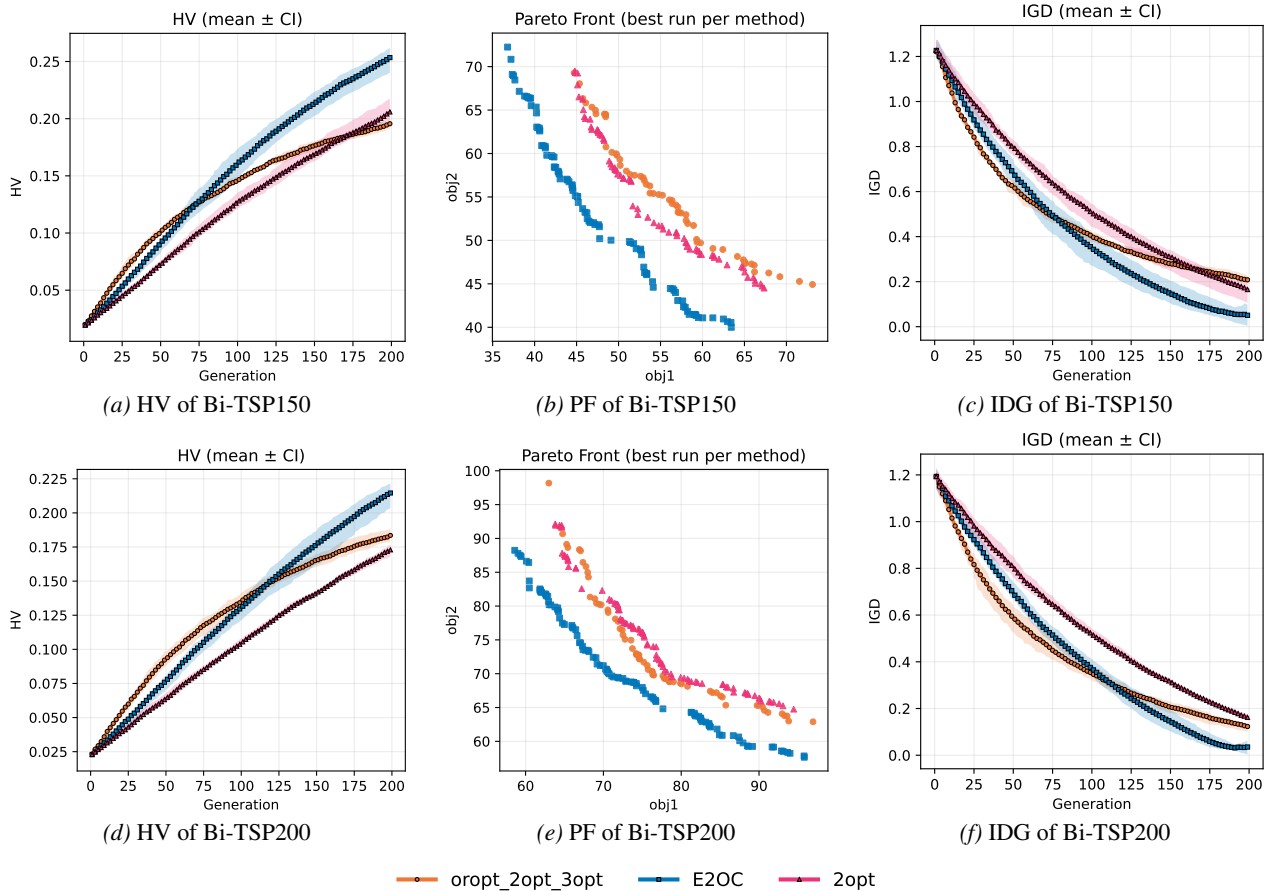

*Figure 14.* HV, IDG and PF performance of different operators in solving Bi-TSPs with different scales.

as the problem scale increases, yielding a significant scale-up dividend; for instance, in Bi-CVRP100 and Bi-KP200, E2OC achieves remarkable RI of 80.59% and 41.64%, respectively, far exceeding the EOH baseline.

To further demonstrate the generalizability of the proposed method, Table 12 presents a cross-scale training-and-testing validation on Bi-CVRP, where operators are trained on scales of 20, 50, and 100, and subsequently cross-evaluated on all sizes. The experimental results reveal that E2OC effectively mitigates small-scale overfitting and exhibits robust cross-scale generalization; when trained on Instance-20 but tested on Instance-100, E2OC delivers a superior HV of **0.0602** compared to 0.0440 for EOH. This proves that by searching within the high-level semantic thought space via MCTS, E2OC successfully captures algorithmic invariants, thereby providing superior robustness for large-scale deployments while effectively supporting low-cost training on smaller instances.

*Table 12.* Comparison of different training sets in Bi-CVRP.

| Training inst. | Test inst. | EOH | E2OC |
|---|---|---|---|
| | 20 | 0.4035±0.0563 | 0.3910±0.0529 |
| 20 | 50 | 0.1928±0.0349 | 0.2230±0.0478 |
| | 100 | 0.0440±0.0188 | 0.0602±0.0424 |
| | 20 | 0.3947±0.0525 | 0.4009±0.0567 |
| 50 | 50 | 0.2200±0.0452 | 0.2473±0.0647 |
| | 100 | 0.0652±0.0473 | 0.0673±0.0515 |
| | 20 | 0.3617±0.0437 | 0.3910±0.0529 |
| 100 | 50 | 0.1926±0.0313 | 0.2418±0.0634 |

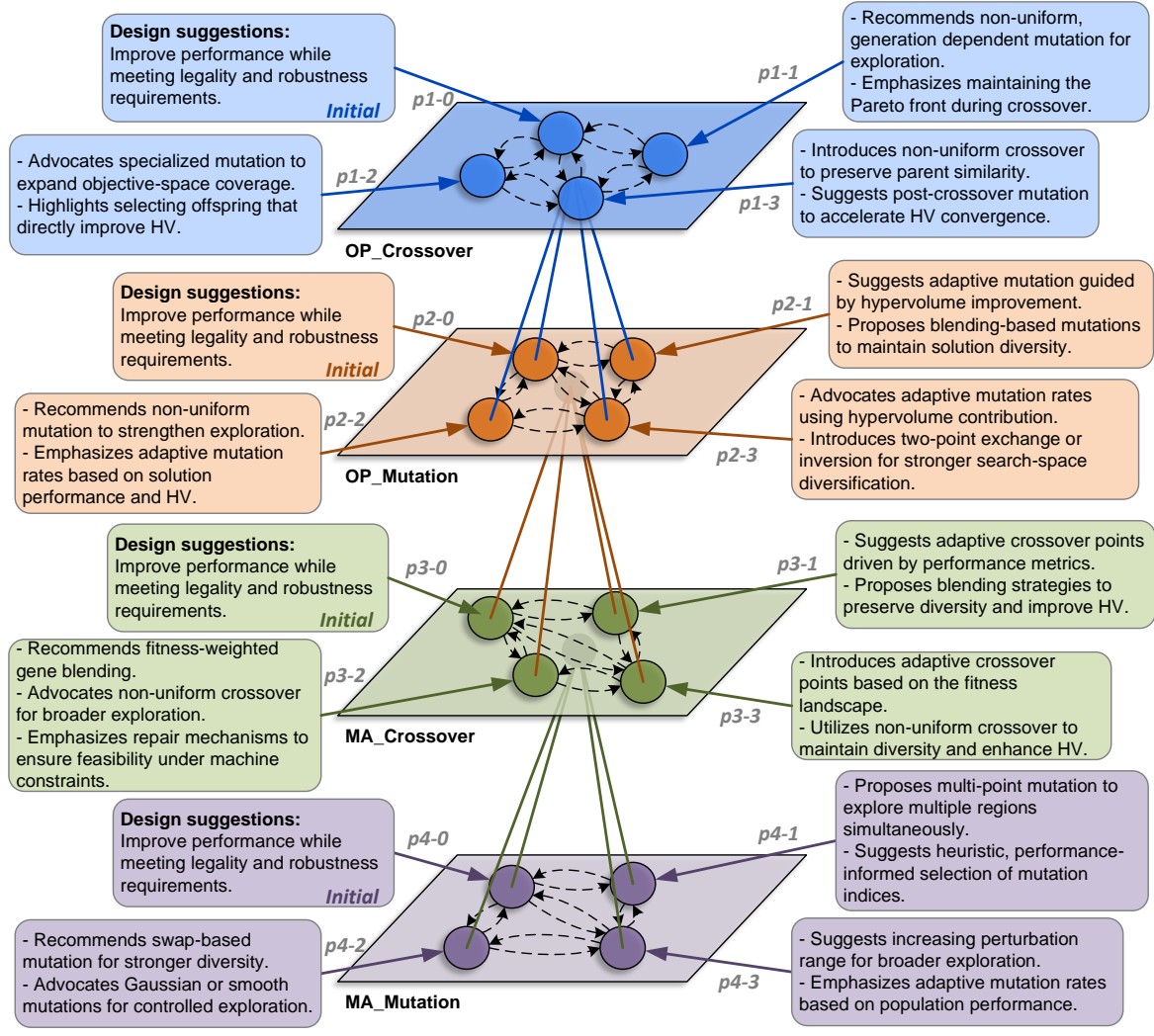

*Figure 15.* Example of a language space constructed by E2OC in Bi-FJSP

## J.6. Interpretability Analysis

During the warm-start phase, E2OC constructs a language space of design thoughts for each operator, composed of multiple improvement suggestions. Similar to multi-operator combinations, these design thoughts exhibit complex and hard-to-quantify coupling relationships. In E2OC, the number of initial added prompts is controlled by the parameter $AP$. A larger $AP$ yields a richer set of operator design thoughts, but also enlarges the combinatorial design space, thereby requiring more computational resources to identify effective design strategies, i.e., optimal combinations of thoughts. This results in an inherent trade-off between search cost and design space expressiveness. To investigate this effect, we conduct a sensitivity analysis on $AP$, as reported in Table 8.

Moreover, different operator design thoughts often possess implicit coupling relations, such as mutual reinforcement, competition, or redundancy. The progressive search of design thought combinations via MCTS in E2OC can be interpreted as an attempt to quantify and exploit these latent interactions through performance-driven evaluation. Taking the Bi-FJSP as an example, we analyze the multi-domain operator design thought space generated by E2OC with $AP = 3$ and NSGA-II as the warm-start baseline. For the FJSP setting, this space includes design thoughts associated with operators acting on different coding regions of the chromosome, namely **operation crossover** and **operation mutation** operators, as well as **machine crossover** and **machine mutation** operators, and it has been distinguished by different colors in Figure 15.

In E2OC, initial design thoughts, code templates, and semantic descriptions are required for each operator prior to the

*Table 13.* The focus of different operator design suggestions in the language space constructed by E2OC in Bi-FJSP.

| Operator | Notation | Focus |
|---|---|---|
| Operation_Crossover | $p1$-0 | Predefined: Constraint- and robustness-first performance optimization |
| | $p1$-1 | Pareto preservation and generation-aware exploration |
| | $p1$-2 | Hypervolume-driven offspring selection |
| | $p1$-3 | Parent-proximity control and convergence |
| Operation_Mutation | $p2$-0 | Predefined: Constraint- and robustness-first performance optimization |
| | $p2$-1 | HV-aware adaptivity and diversity preservation |
| | $p2$-2 | Exploration control via mutation rate adaptation |
| | $p2$-3 | Structural diversification driven by HV contribution |
| Machine_Crossover | $p3$-0 | Predefined: Constraint- and robustness-first performance optimization |
| | $p3$-1 | Performance-aware crossover and diversity maintenance |
| | $p3$-2 | Fitness-weighted recombination and feasibility handling |
| | $p3$-3 | Fitness-landscape-guided exploration |
| Machine_Mutation | $p4$-0 | Predefined: Constraint- and robustness-first performance optimization |
| | $p4$-1 | Multi-point exploration with experience guidance |
| | $p4$-2 | Structural diversity through value replacement |
| | $p4$-3 | Adaptivity and exploration intensity control |

warm-start phase. For multi-objective optimization operators, we adopt a minimal and generic initialization principle: ensuring legality and robustness while pursuing performance improvement. The design thoughts of other operators are automatically derived by the LLM through advantage analysis of high-performing operators observed during warm-start, resulting in structured improvement suggestions. As summarized in Table 13, these suggestions emphasize different design focuses across operators. The combinations of such heterogeneous focuses constitute the diversity of design strategies and form the basis of interpretability in E2OC.

*Figure 16.* Machine crossover operators designed by EoH and E2OC on FJSP.

## J.7. Designed Operators

This section compares the FJSP crossover operators generated by E2OC with those obtained from manual design and EoH, focusing on their design motivations and implementation characteristics. Although EoH and E2OC both employ LLMs to assist operator construction, they differ in how design knowledge is represented and incorporated into the final operators. The specific implementations of these operators are shown in Figure 17 and Figure 16. The performance differences are shown in Table 3.

```python
def Operation_Crossover(parent1: np.ndarray, parent2: np.ndarray, n_vars:
int) -> Tuple[np.ndarray, np.ndarray]:

    # Implement operation priority crossover
    if n_vars < 2:
        # Direct copy for extremely small length to avoid randint(1,1) error
        return parent1.copy(), parent2.copy()

    cp = np.random.randint(1, n_vars)

    child1 = np.concatenate([parent1[:cp], parent2[cp:]])
    child2 = np.concatenate([parent2[:cp], parent1[cp:]])

    return child1, child2
                                                    Initial
```

```python
def Operation_Crossover(parent1: np.ndarray, parent2: np.ndarray, n_vars:
int) -> Tuple[np.ndarray, np.ndarray]:

    if parent1.shape != (n_vars,) or parent2.shape != (n_vars,):
        return parent1.copy(), parent2.copy()

    try:
        point1, point2 = np.sort(np.random.choice(range(1, n_vars), size=2,
replace=False))

        child1 = np.concatenate([parent1[:point1], parent2[point1:point2],
parent1[point2:]])
        child2 = np.concatenate([parent2[:point1], parent1[point1:point2],
parent2[point2:]])

        child1 = np.clip(child1, 0, 1)
        child2 = np.clip(child2, 0, 1)

        return child1, child2

    except:
        return parent1.copy(), parent2.copy()
                                                    EOH
```

```python
def Operation_Crossover(parent1: np.ndarray, parent2: np.ndarray, n_vars: int) ->
Tuple[np.ndarray, np.ndarray]:

    # Placeholder for fitness-based alpha calculation
    def calculate_alpha(fitness1, fitness2):
        return 0.5 + 0.2 * (fitness1 - fitness2) / (abs(fitness1) + abs(fitness2) + 1e-10)

    # Assume fitness is the negative of the objective function value for HV maximization
    # Replace with actual fitness evaluation logic
    fitness1 = -np.sum(parent1)
    fitness2 = -np.sum(parent2)
    alpha = calculate_alpha(fitness1, fitness2)
    r1 = np.linalg.norm(parent1)
    r2 = np.linalg.norm(parent2)
    theta1 = np.arctan2(parent1[1:], parent1[:-1]) if n_vars > 1 else np.array([])
    theta2 = np.arctan2(parent2[1:], parent2[:-1]) if n_vars > 1 else np.array([])

    # Cyclic rotation of angles
    theta_child1 = np.roll(theta1, 1) * (1 - alpha) + theta2 * alpha
    theta_child2 = np.roll(theta2, 1) * (1 - alpha) + theta1 * alpha

    # Blend radii
    r_child1 = alpha * r1 + (1 - alpha) * r2
    r_child2 = (1 - alpha) * r1 + alpha * r2

    # Convert back to Cartesian coordinates
    def polar_to_cartesian(r, angles):
        if n_vars == 1:
            return np.array([r])
        elif n_vars == 2:
            return np.array([r * np.cos(angles[0]), r * np.sin(angles[0])])
        else:
            coords = np.zeros(n_vars)
            coords[0] = r
            for i in range(1, n_vars - 1):
                coords[i] = r * np.prod(np.cos(angles[:i])) * np.sin(angles[i - 1])
            coords[-1] = r * np.prod(np.cos(angles[: n_vars - 2]))
            return coords

    child1 = polar_to_cartesian(r_child1, theta_child1)
    child2 = polar_to_cartesian(r_child2, theta_child2)

    return child1, child2
                                                    E2OC
```

*Figure 17.* Operation crossover operators designed by EoH and E2OC on FJSP.

For the operation-sequence crossover, shown in Figure 17, manually designed and EoH-designed operators remain within positional segment exchange, differing mainly in recombination granularity. The E2OC-designed operator instead reformulates crossover in a continuous geometric space, where offspring are generated via fitness-aware interpolation between parent representations. This design reflects E2OC's ability to explore couplings between operators and evaluation feedback, as the operator form is shaped not only by structural validity but also by how effectively it cooperates with other operators under repeated evaluation. Such non-positional recombination strategies are difficult to obtain through isolated heuristic evolution but naturally arise under E2OC's multi-operator co-evolution framework.

For the machine-selection crossover, shown in Figure 16, the manually designed operator applies a single-point positional recombination, which is simple and problem-agnostic but ignores solution-level feedback. The EoH-designed operator emphasizes structural preservation by explicitly retaining identical decision components and introducing random exchanges only on divergent positions, reflecting a design bias toward stable inheritance within a single-operator evolution setting. In contrast, the E2OC-designed operator incorporates objective-space distance into the crossover decision, adaptively adjusting recombination strength according to parent similarity. This behavior emerges from E2OC's progressive design strategy search, where different recombination principles are explored and selected through operator rotation under performance feedback, rather than being fixed a priori.

Overall, the observed performance advantages of E2OC-designed operators stem not from increased operator complexity, but from the systematic exploration of design strategies and their interactions enabled by operator rotation evolution. By allowing multiple design thoughts to be decoupled, recombined, and empirically validated across operators, E2OC produces operators that exhibit more adaptive and context-aware recombination behaviors, leading to more robust performance on

FJSP instances.

## K. Limitations

Compared to existing single-heuristic design methods, E2OC has achieved promising performance in MOEAs through the co-evolution of design strategies and executable codes for multiple operators. However, this approach still presents several limitations:

**High Evaluation Cost of Design Thoughts Search.**   Expanding the search for design thoughts significantly increases computational demands. This results in substantial resource consumption during evaluation, presenting a major challenge for achieving efficient algorithm iteration in practice.

**Dependence on Domain Knowledge and Capability Boundaries of LLMs.**   E2OC heavily relies on domain knowledge provided by LLMs, as its core mechanism involves the combinatorial search of algorithmic design knowledge.  The effectiveness of this approach, however, is constrained by the quality of the domain knowledge, which is inherently dependent on the capabilities of the underlying large language model.

**Experience Driven Design Strategy Evaluation.**   The present work employs an experience driven paradigm, where design strategies are selected and optimized based on empirical performance outcomes rather than formal semantic verification. This approach prioritizes practical effectiveness, acknowledging that semantic adherence remains an open challenge in automated design.  While the framework provides explanatory power at the strategic level, it does not guarantee literal implementation fidelity. Future work may explore incorporating semantic consistency validation to enhance reliability.

These challenges highlight key issues that mainstream LLM-based AHD methods need to address in future research.

