# OpenReview forum: "Evolving Interdependent Operators with Large Language Models for Multi-Objective Combinatorial Optimization"
_ICML.cc/2026/Conference — ICML 2026 regular_

### Official Review · Reviewer_BT3r · 2026-03-06

**Soundness:** 3
**Presentation:** 3
**Significance:** 2
**Originality:** 2
**Overall Recommendation:** 3
**Confidence:** 5

**Summary:**

This paper proposes E2OC, a framework for automatically designing interdependent operator combinations for multi-objective evolutionary algorithms using LLM-guided search. The authors argue that most existing automated heuristic design approaches focus on evolving individual operators independently, even though operators in evolutionary algorithms interact in complex ways.

To address this, the paper proposes evolving sets of operators simultaneously using a combination of:
•	LLM-generated “design thoughts” [a rather cringe-worthy term that seems more LinkedIn than ICML!]

•	Monte Carlo Tree Search to explore combinations of these thoughts

•	an operator rotation mechanism that iteratively replaces operators and evaluates performance.

The resulting framework is evaluated on two classical combinatorial optimization problems (multi-objective flexible job shop scheduling and multi-objective TSP), where the automatically designed operator combinations outperform expert-designed baselines and several existing automated heuristic design frameworks

**Compliance With Llm Reviewing Policy:**

Affirmed.

**Final Justification:**

The rebuttal improves the paper in several respects: however, my main reservations remain. In particular, the methodological contribution still seems largely integrative, and more importantly, there is still a mismatch between the paper’s central semantic-level framing and what is actually verified in the system: operators are ultimately selected based on executability and downstream performance rather than direct validation that the generated code faithfully implements the intended design thoughts. Since this semantic adherence issue is closely tied to the paper’s core claim, and the empirical scope remains somewhat limited, I maintain my original recommendation.

**Key Questions For Authors:**

1. How general is the operator-interaction framework?
The experiments focus on two specific combinatorial optimisation problems. Do you expect the same design-strategy search process to work for other MOEA domains? Some discussion of how general this framework is intended to be would be nice.
2. Sensitivity to design thought space initialization
The search space is constructed from design thoughts extracted during the warm-start stage. How sensitive are the results to the quality of these initial thoughts? For example, if the initial LLM suggestions were poor, would the search still converge to strong operator combinations? Yes, they’d get stronger, but strong enough to make it worth it?
3. Scalability of the framework
As the number of operators increases, the number of possible combinations grows quickly. How well does the MCTS search scale in this setting? At some point it’ll collapse, so it would be nice to know (or estimate) when that could happen.
4. Understanding operator interactions
Does the framework provide any insights into why particular operator combinations perform well, or does it primarily identify them through empirical search? I suspect the latter

**Limitations:**

You do a nice job of acknowledging the limitations of your approach:
•	 Small set of benchmark problems.
•	The system depends heavily on LLM-generated suggestions, whose quality may vary across models.
•	The design search space is influenced by the initial warm-start operator pool.
•	The approach focuses on empirical discovery rather than theoretical understanding of operator interactions.
These are reasonable limitations for this type of work, but they are worth keeping in mind when interpreting the results.

**Strengths And Weaknesses:**

What I liked

Clear motivation

Clear motivation well explained. The paper correctly identifies that most automated heuristic design approaches evolve individual operators independently, even though the performance of MOEAs depends heavily on how operators interact. Fair.

Reasonably structured framework

The framework itself is logically structured (warm-start to design thought space to  MCTS search to operator rotation evaluation). The idea of searching over “design thoughts” rather than directly over code is also conceptually interesting. This gives a coherent pipeline.

Strong empirical improvements

Across the epxeriments,  the results show consistent improvements over baselines operators and existing automated heuristic design methods.While the absolute magnitude of the results won’t change the world, they are consistent and believable

Good presentation

A few too many buzzwords – “design thoughts”, I’m talking to you! – but the paper is well structured and the diagrams are nice.

What I liked less

Incremental methodological novelty

While the overall framing is interesting, most of the individual components are already familiar:
•	LLM-based heuristic design
•	Monte Carlo Tree Search for strategy exploration
•	evolutionary improvement of heuristics

So the novelty lies mainly in combining these elements for multi-operator design rather than introducing fundamentally new methodology. Okay, nice, but is this just a groovy framework integration than a fundamentally new contribution?

The “design thoughts” abstraction is quite vague

A central concept in the framework is the use of “design thoughts” extracted from operators and used to guide the search process. In practice, these appear to be LLM-generated textual suggestions about operator modifications.
While this abstraction sounds appealing, the paper never fully clarifies:
•	how meaningful these thoughts actually are
•	how strongly they influence the search
•	whether they provide more value than simply sampling candidate operators.
As a result, the system sometimes feels like a sophisticated wrapper around heuristic search over operator implementations

Limited experimental scope

The evaluation is restricted to two types of combinatorial optimization problems:
•	flexible job shop scheduling
•	multi-objective TSP.
While these are standard benchmarks, they represent only a small portion of the broader space of multi-objective optimization problems. It is therefore unclear how well the proposed framework generalizes to other problem classes

The mechanism of operator interaction is unclear

The paper argues that modeling operator interactions is critical, but the framework ultimately identifies good operator combinations through empirical search rather than explicit modeling of interaction effects.
As a result, the system demonstrates that certain operator combinations work well, but it provides limited insight into WHY those combinations perform better.

---

> ### Author Rebuttal · Authors · 2026-03-31
>
> Your comments have provided us with many valuable insights, and we are very grateful. Our point-by-point response is as follows:
>
> **1. Methodological novelty (Weak.1):**
> We appreciate the reviewer’s perspective. E2OC’s core novelty is the Semantic Strategy Co-evolution paradigm, which directly addresses the unaddressed challenge of co-designing interdependent operators in MOEAs.
>
> Current LLM-AHD methods are designed for single-operator optimization and fail to consider operator couplings, as shown by the poor performance of sequential single-heuristic design in Table 2. E2OC introduces a co-evolutionary loop between a semantic strategy space(composed of design thoughts) and the executable code space, shifting optimization from syntax-level changes to intent-level strategy.
>
> This paradigm is empirically validated. Tables 2 and 5 compare E2OC against multiple multi-operator co-design frameworks and MCTS variants. Only the E2OC achieves superior performance, demonstrating that its specific integration of warm-start(to curate a high-quality initial strategy space) and progressive MCTS search is essential, not merely the use of MCTS or LLMs.
>
> The work provides the key insight that effective co-design requires prioritized search in a bounded, semantically-rich strategy space. This enables the emergence of functionally complementary operators, a contribution whose depth surpasses simple framework integration.
>
> **2. Design Thoughts Abstraction (Weak.2):**
> We appreciate the reviewer’s questions regarding the design thoughts. Design thoughts are concrete textual improvement suggestions that extend the “semantic idea” concept from EoH. Extracted using the template in Fig. 4, their impact is measurable: the best strategy achieves an HV of 0.1819 while the worst yields 0.1677 under the same budget (Fig. 15). This substantial performance gap, difficult for human engineers to achieve under the same constraints, confirms that the specific semantic content of a design thought directly determines the optimization capability of the resulting algorithm.
>
> Design thoughts are the primary driver of the search. E2OC’s MCTS navigates the combinatorial space of these thoughts, using the resulting code’s HV as the sole reward signal. This relationship is analyzed in supplementary ablation experiments(Table S5): replacing any operator in the final combination with its initial version degrades performance, showing that the synergistic operator set is a direct outcome of the selected thoughts.
>
> Moreover, searching this curated semantic space is more effective than sampling operators directly in unbounded code space. Frameworks that sample without semantic guidance (CD, UCB) or sample thoughts dynamically (MCTS_Sample, MCTS_Tuple) are consistently outperformed. E2OC’s bounded semantic search is therefore not a wrapper for heuristic sampling, but a strategy-first paradigm that delivers greater efficiency.
>
> **3. Limited experimental scope (Weak.3 & KQ.1):**
> We have further supplemented more validation of generalization and overfitting. Due to serious rebuttal length limitations, we have provided a comprehensive response in ***Reviewer x4Ld (Point 2)***, which you are kindly requested to review. Please refer to that response and feel free to ask further questions.
>
> **4. Sensitivity to Design Thought Space (KQ.2):**
> Due to space limitations, we have addressed the sensitivity analysis related to the LLM and the design thought space in our response to ***Reviewer x4Ld (Point 6)***.
>
> **5. Scalability with more operators (KQ.3):**
> E2OC effectively manages scalability through a bounded-search-space design. We have provided a more detailed analysis of the changes in tree structure and growth rate in the response to ***Reviewer x4Ld (Point 4)***.
>
> Collapse occurs when the space is too vast for MCTS to maintain adequate sampling density for reliable UCB estimates. For typical MOEA tasks ($K$=3/4), E2OC's bounded design maintains high efficiency. For $K>10$, scalability remains manageable by tuning $AP$ and $iter\_{out}$, preventing the combinatorial explosion typical of the unbounded approaches seen in MCTS_OC and MCTS_Tuple.
>
> **6. Operator Interactions (KQ.4):**
> Yes, it primarily searches for interactions through empirical search. We have provided a detailed unified explanation in the response to ***Reviewer x4Ld (Point 5)***, which addresses this question.
>
> **7. Limitations:**
> Thank you once again for your understanding of the limitations in our work. Given the state of existing LLM-based theoretical tools and AHD techniques, E2OC primarily adopts an experience-driven or data-driven exploratory perspective​ in co-designing both the executable code and design strategy of interdependent operators. In the revised manuscript, we will adjust the description of the experimental results accordingly.
>
> All newly added tables can be found in ***Reviewer 3ag4 Supplementary*** or link.
> We can continue to communicate if you have any questions. Thank you.

---

> > ### Author Rebuttal · Reviewer_BT3r · 2026-04-05
> >
> > The rebuttal addresses several of my concerns reasonably well.
> >
> > In particular, the role of the “design thought” space is now clearer, with some evidence that it contributes beyond simple prompt variation. The discussion of scalability is also improved, and the bounded search-space argument makes the use of MCTS more convincing.
> >
> > That said, two issues still feel only partially resolved. First, the methodological novelty seems to come mainly from integrating existing components rather than introducing a fundamentally new idea. Second, while there is some qualitative evidence of operator synergy, the framework still identifies effective combinations largely through empirical search, with limited insight into why they work.
> >
> > Overall, the rebuttal strengthens the paper, but I still have some reservations around novelty and interpretability.

---

> > > ### Author Response · Authors · 2026-04-06
> > >
> > > We sincerely thank the reviewer's detailed review and positive feedback. We are very pleased that you find most concerns addressed and acknowledge the clarifications regarding the design thoughts and scalability. To address the remaining reservations regarding novelty and interpretability, we provide the following clarification to further articulate the core contributions of this work:
> > >
> > > ### **On Novelty**
> > > E2OC is not a simple integration of existing components like MCTS, and its fundamental contribution lies in proposing a systematic new paradigm for a previously unaddressed problem: **how to automatically and cooperatively design interdependent operators for MOEAs**. Its novelty is demonstrated at three levels:
> > >
> > > 1. **New Problem Definition**: Unlike single-heuristic AHD methods, we explicitly model the interdependencies between multiple operators as a MDP (Appendix C). This formalizes the co-design challenge as a distinct optimization problem.
> > >
> > > 2. **New Methodological Paradigm**: E2OC introduces **Semantic Strategy Co-evolution**, creating a closed-loop co-evolution between the space of semantic design strategy and code, including:
> > >     - We **operationalize design thoughts into searchable, composable semantic units**, extending and deepening​ the semantic‑idea concept introduced in EoH and MCTS‑AHD.
> > >     - MCTS is applied to the semantic strategy space rather than raw code. This enables efficient exploration of strategy-level interplay.
> > >     - Rotation evaluation simulates dynamic interactions between operators, providing MCTS with performance-based feedback to assess couplings.
> > >
> > > 3. **Empirical Distinction from Simple Integration**: Tables 2 and 5 prove that E2OC significantly outperforms baseline schemes that simply combine existing components (e.g., CD, UCB, LLM). This confirms that the success stems from a non-trivial architecture—specifically the integration of strategy-space bounding, progressive MCTS, and rotation evaluation—rather than a mere assembly of parts.
> > >
> > > The innovation lies in identifying a new problem, proposing a new paradigm, and designing a specific architecture that outperforms all simple integration variants. Integrating technologies at the right level to solve new problems goes far beyond the scope of component assembly, and **has important implications for other multi-component co-designs**.
> > >
> > > ### **On Interpretability**
> > > While quantitative analysis of the coupling between design thoughts and code remains challenging, E2OC moves beyond black-box search by providing multi-level insights into operator synergy. Its unique design enables traceable analysis of how specific strategy combinations drive performance gains.
> > >
> > > - **Semantic Guidance**: Unlike FunSearch, which directly optimizes code, or EoH and MCTS-AHD, which add idea labels, E2OC uses structured design thoughts as **human-understandable anchors**. This ensures evolution is guided by specific, interpretable improvements rather than generic stochastic prompting.
> > > - **Traceable Evolutionary Paths**: Our progressive strategy search provides a clear trajectory for thought combinations. The MCTS scores serve as explicit selection criteria, **quantifying the interaction and synergy between high-level strategies**.
> > > - Formal Theoretical Foundation: We provide a rigorous framework for **multi-operator dependencies via a MDP**. This structures the co-design challenge as a sequential decision problem, laying the groundwork for systematic analysis.
> > > - Causal Experimental Validation: Systematic ablation studies (Tables 3 & 5) prove that semantic strategy search and rotation evaluation are functionally necessary. This demonstrates why E2OC is effective: **it explicitly addresses the coupling dependencies that independent design ignores**.
> > > - Operator Implementation Analysis: In-depth analysis of optimal operators (Table 11, Figs. 16-17) reveals how high-level intents translate into complementary code logic. This confirms that synergy arises from a clear division of labor at the implementation level, providing a tangible engineering basis for the system‘s internal workings.
> > >
> > > While concrete semantic mapping presents a deeper theoretical challenge than the interpretability issues faced by early systems like AlphaGo, E2OC pushes interpretability to its current practical limits. **It validates a new paradigm of co-optimization in the thought space through empirical traceability and mechanistic feedback**.
> > >
> > > We sincerely appreciate your constructive feedback, which guided the additional analysis and evidence provided in this work. We will incorporate more detailed descriptions in the final manuscript. Given the current **Weak Reject**​ recommendation, **we would be grateful if you would take another look at the contribution of this work** to the paradigm of co‑optimizing design strategies and code for interdependent operators in MOEAs. We believe the refinements made have meaningfully strengthened the manuscript.
> > >
> > > Once again express our gratitude to you.

---

### Official Review · Reviewer_3ag4 · 2026-03-12

**Soundness:** 4
**Presentation:** 4
**Significance:** 3
**Originality:** 3
**Overall Recommendation:** 5
**Confidence:** 4

**Summary:**

This paper proposes a new framework named E2OC, which utilizes LLMs to automatically and co-design interdependent operator combinations for MOEAs. While most current LLM-based AHD methods are limited to optimizing individual operators in isolation, E2OC overcomes this limitation by co-evolving design strategies alongside executable code. Its core mechanism employs MCTS to explore optimal multi-operator design strategies within a semantic-level design thought space, followed by an efficient evaluation through the operator rotation mechanism.
To validate the effectiveness of E2OC, the authors designed a series of comprehensive experiments on multi-objective TSP and FJSP. These include: analyses of different LLMs and parameter settings, ablation studies, comparisons with mainstream MOEAs, comparisons with SOTA AHD methods, evaluations of different MCTS variants, and tests of its continuous optimization potential. The experimental results robustly support the effectiveness, superiority capability of E2OC.

**Compliance With Llm Reviewing Policy:**

Affirmed.

**Final Justification:**

My concerns are addressed. I maintain my recommendation to accept.

**Key Questions For Authors:**

1. Can E2OC be extended to optimize multi-operator systems for single-objective problems?
2. How can the diversity of the initial design knowledge space be ensured?
3. The initial multi-domain design knowledge space might lead the multi-operator semantic combinations into local optima. Would dynamically reflecting upon and updating this knowledge space during optimization be more effective?

**Limitations:**

yes

**Strengths And Weaknesses:**

Strengths
1. Well-Defined Problem: Clearly identifies the limitations of existing methods, which primarily optimize operators in isolation, formulates the challenge of co-designing interdependent operator combinations in MOEAs, and defines it as a MDP to model their implicit coupling.
2. Novel and Systematic Methodology: Proposes a new paradigm of co-evolving design strategies with executable code. E2OC integrates: warm-start/semantic space construction, progressive MCTS-based strategy search, and operator rotation evolution, essentially creating an evolutionary system for meta-optimization at the semantic level.
3. High-level Semantic Combinatorial Optimization: Fills a gap in LLM-based AHD by introducing a new multi-heuristic design paradigm. Specifically, E2OC performs search and combination within an unstructured semantic space of language design thoughts to discover design strategies that guide the generation of high-performance code.
4. Comprehensive Experiments: Features extensive evaluations against many SOTA LLM-based AHD methods and expert-designed operators, alongside thorough analyses of its own parameters, components, and various MCTS-based multi-operator search variants, providing valuable guidance for the field.

Weaknesses
1. The paper describes E2OC as an "offline design" and compares it to other SOTA AHD methods under controlled total cost (total LLM calls or code evaluations). However, without a conclusive summary, readers may find it difficult to understand the advantages of E2OC.
2. The multi-domain design knowledge space constructed during the warm-start phase is limited in size. Although the "initial number of added prompts" was analyzed, it may fall into a local optimum in more complex scenarios.

---

> ### Author Rebuttal · Authors · 2026-03-31
>
> Thank you for the insightful and thorough review. Below are our point-by-point responses:
>
> **1. Conclusive Summary of Cost Advantage (Weak.1):**
> We will add a clear summary in the revised manuscript. The cost analysis in ***Reviewer x4Ld (Point 1)*** and Table S1 provides a direct comparison of API calls and evaluations. Following recent LLM‑AHD work (EoH, ReEvo, MCTS‑AHD), we enforce strict parity in total evaluation counts. Results show that E2OC, by constructing a high‑quality initial design‑thought space through warm‑start, prevents wasteful exploration in poor strategic regions and delivers superior performance under the same budget.
>
> **2. Local Optima in Limited Initial Design Space (Weak.2):**
> We acknowledge that the initial design-thought space is bounded relative to the full algorithm space, which could theoretically restrict exploration. However, E2OC includes a built‑in mechanism to overcome this limitation. Our iterative optimization experiments (Table 5) show that when the best operator combination from one E2OC run is used as the initial template for a subsequent run, even better design strategies and higher performance are achieved. This demonstrates that E2OC supports sustained, open‑ended improvement across multiple design cycles.
>
> **3. Extensibility to Single-Objective Problems (KQ.1):**
> E2OC can be extended to single-objective optimization. The core challenge in multi-objective problems is the trade-off between conflicting goals, which creates complex operator interdependencies. For single-objective problems, this complexity is reduced. Adapting E2OC primarily involves switching the internal evaluator to a single-objective evolutionary algorithm. The warm-start and progressive search mechanisms of E2OC do not face new fundamental challenges in this setting. We plan to demonstrate this in future work and will open-source the framework for broader application.
>
> **4. Diversity of the Initial Design Knowledge Space (KQ.2):**
> E2OC derives initial design-thought diversity by extracting "improvement suggestions" from operators with varied performance scores during the warm-start phase. By semantically analyzing different operators, the framework naturally gathers a diverse set of strategic intents without requiring external expert knowledge. To further enhance this diversity, we identify two primary methods:
> - Enriching Warm-Start Search: Using more sophisticated mechanisms to increase initial operator diversity.
> - Incorporating Expert Knowledge: Seeding the design-thought space with expert-provided ideas alongside LLM-generated analysis.
>
> **5. Dynamic Reflection vs. Fixed Initial Space (KQ.3):**
> We extensively experimented with dynamic design-thought updating mechanisms, such as MCTS_Sample​ (dynamically generating new thoughts up to a limit) and MCTS_Tuple​ (unbounded sampling of new thought combinations).
>
> Results in Table 5 show that E2OC (with a fixed, bounded initial thought space) outperforms these dynamic variants. This indicates that unrestricted exploration of new thoughts can harm performance by overshadowing valuable historical knowledge. E2OC strikes a better balance between exploration and exploitation.
>
> This conclusion is further supported by comparisons with other reflective frameworks (e.g., LLM, Win-UCB in Table 2), where E2OC consistently achieves superior performance. Thus, while dynamically updating thoughts can aid in escaping local optima, E2OC’s strategy of refining a curated, bounded initial space proves more effective overall.
>
> ## Supplementary
>
> **1. Table S1. Budget comparison.**
>
> |Para.|EOH|CD|UCB|MCTS_OC|MCTS-Sample|MCTS-Tuple|E2OC|
> |:---|:---|:---|:---|:---|:---|:---|:---|
> |Warm-start|-|-|1|1|1|1|1|1|
> |Outer iteration ($iter_{out}$)|-|-|30|30|30|30|30|
> |Number of initial operators ($K$)|4|4|4|4|4|4|4|
> |Number of initial added prompts ($AP$)|-|-|-|-|3|-|3|
> |Middle iteration ($iter_{middle}$)|-|154|5|5|5|5|5|
> |Inner iteration|-|-|-|-|-|-|-|
> |Operator population size|-|20|10|10|10|10|10|
> |Max sampling number ($sam_{max}$)|3875|25|25|25|25|25|25|
> |Number of evaluations|15500|15500|15500|15500|15500|15500|15500|
> |Number of design thoughts|0|0|0|0|12|15512|12|
> |Total|15500|15500|15500|15500|15512|31012|15512|
>
> **2. Please refer to the anonymous link (https://anonymous.4open.science/r/Anoymous_code-2CCE/11963_rebuttal_appendix.md) for other new experimental results:**
> - Table S2. Evaluation parameter settings.
> - Table S3. Comparison on more MOPs.
> - Table S4. Comparison of different training sets.
> - Table S5. The ablation results of the final design thought combination.
> - Table S6. Statistical significance testing.
>
> If you have any further questions, please feel free to ask. Thank you.

---

> > ### Author Rebuttal · Reviewer_3ag4 · 2026-04-01
> >
> > I have reviewed the authors' rebuttal, which addresses my questions point by point. It satisfactorily resolves the main concerns regarding the sensitivity of the thought space and its generalization . The responses on budget and the analysis of interdependence are also convincing. This is solid work, therefore, I maintain my original recommendation to accept.

---

### Official Review · Reviewer_UaRm · 2026-03-12

**Soundness:** 2
**Presentation:** 2
**Significance:** 3
**Originality:** 3
**Overall Recommendation:** 3
**Confidence:** 5

**Summary:**

This paper introduces E2OC, an automated heuristic design (AHD) framework that co-evolves interdependent search operators for multi-objective evolutionary algorithms (MOEAs). Prior LLM-driven AHD methods largely optimize operators in isolation; E2OC attempts to model operator synergies by formalizing a "language space" of textual design strategies. It employs Monte Carlo Tree Search (MCTS) to navigate combinations of these semantic strategies, paired with a coordinate-descent-style operator rotation mechanism to iteratively refine executable code. Evaluated on bi- and tri-objective Flexible Job Shop Scheduling (FJSP) and Traveling Salesman Problems (TSP), E2OC demonstrates improvements in Hypervolume (HV) and Inverted Generational Distance (IGD) over expert-designed heuristics and sequential LLM baselines.

**Compliance With Llm Reviewing Policy:**

Affirmed.

**Final Justification:**

The semantic consistency/adherence should be critical if the proposed framework "achieves the co-evolution of design strategies and execution code", as stated in the abstract of the paper. However, I see a mismatch between such claim and the present paper. So I maintain my score.

**Key Questions For Authors:**

1. How exactly does the prompt generator enforce that the LLM's output code accurately adheres to the selected "design thought"? What is the failure rate (syntax or logical errors) during the rotation updates, and how are these failures masked or penalized in the MCTS reward signal?

2. Can you provide a detailed, quantitative breakdown of the computational budget (total LLM tokens, API calls, and environment rollouts) for E2OC versus the baselines (e.g., EoH, Win-UCB)?

3. How sensitive is E2OC to the operator rotation ordering and the stopping criteria? Did you evaluate dynamic or bandit-based scheduling instead of a fixed sequential rotation?

4. Can you provide statistical significance tests (e.g., paired Wilcoxon tests) ?

5. How does E2OC position itself against recent multi-component co-design frameworks like LaGO or G-LNS?

**Limitations:**

Yes.

**Strengths And Weaknesses:**

### Strengths
- The shift from single-operator AHD to explicitly modeling multi-operator interdependencies addresses a well-known bottleneck in MOEA design.
- The modularity of the operator-rotation mechanism allows seamless integration with existing AHD generators (e.g., EoH, FunSearch, MCTS-AHD), broadening the framework's practical applicability.
- Ablation studies (e.g., Table 5 comparing MCTS variants) effectively isolate the contributions of the framework's components

### Weakness
- Statistical significance testing is missing.
- The framework involves multiple computationally heavy phases (warm-start, thought extraction, MCTS, and rotation-based evaluations). While the paper claims budget parity across baselines, it does not provide a transparent, granular accounting of token usage, LLM API calls, and exact fitness evaluation counts per method.
- The pipeline for translating semantic "design thoughts" into executable, bug-free code is treated as a black box. There is no rigorous explanation of how the framework verifies that the generated code mathematically implements the selected MCTS prompt strategy without semantic drift.
- Generalization claims rest on extremely narrow train/test splits (e.g., training on a single instance like mk15 for FJSP or a single 100-node setting for TSP). This setup risks overfitting to specific topological features of the training instance rather than learning generalized operator synergies.

---

> ### Author Rebuttal · Authors · 2026-03-31
>
> Thank you for the insightful and thorough review. Your comments are crucial for improving the rigor of our work. Below are our point-by-point responses.
>
> **1. Statistical Significance Testing (Weak.1 & KQ.4):**
> Thank you for the suggestion. We have added statistical significance testing to the key experimental data. Furthermore, paired Wilcoxon signed-rank tests ($\alpha=0.05$) confirm that E2OC’s improvements over expert-designed, single-, and multi-heuristic baselines are statistically significant ($p<0.05$) across different instances. These summaries will be added to the revised version.
>
> **2. Budget analysis (Weak.2 & KQ.2):**
> To clarify evaluation parity, we have added details regarding the API call and evaluation budget for different methods. Following LLM-AHD conventions, we enforce a strict parity in total fitness evaluations. E2OC demonstrates superior performance under this same budget, ensuring a fair and rigorous comparison.
>
> **3. Verification of thought-to-Code Mapping (Weak.3):**
> The mapping from semantic "design thoughts" to executable code is indeed a core open challenge. E2OC does not attempt formal mathematical verification. It guides the performance of operator combinations inferred from different design strategies, which is an experience- or data-driven approach.
>
> Within MCTS, this HV acts as the reward signal. The search naturally selects and retains thoughts that yield high-HV code while pruning those leading to low-performance or invalid results. Thus, semantic alignment is enforced by the optimization-driven selection pressure itself. The warm-start further mitigates risks of semantic drift by initializing the search in a curated, high-quality semantic space.
>
> **4. Thought adherence and failure impact (KQ.1):**
> E2OC enforces adherence dynamically through MCTS selection rather than static constraints. The framework retains only design thoughts that produce high-performance code, even when syntax is correct (Table 4). This is empirically validated in Table S5, showing that distinct, surviving design thoughts lead to complementary functionalities, confirming the effective transmission of strategic intent.
>
> Code generation success depends on LLM capability. Using high-capability models like 4o-mini, the success rate reaches 100% (Table 4). For other models, the low failure rate is managed by fault-tolerance mechanism:
>
> During offline design, AHD modules discard non-executable or uncompilable codes. MCTS will backpropagate the best score to the corresponding node. The nodes with inferior operators will have a lower probability of being selected, ensuring their natural elimination from the search tree.
>
> **5. Sensitivity to Operator Rotation (KQ.3):**
> The operator rotation mechanism acts as a meta-evaluator for scoring design strategy. The order and number of rotations determine resource allocation for this evaluation, primarily affecting the precision of the strategy score rather than dictating an absolute best order.
>
> We have compared various ordering strategies within the multi-operator design framework (Table 2): Fixed order (CD), Bandit-based dynamic selection (UCB, Win-UCB), LLM-driven decision (LLM). While Win-UCB performs well during design, the E2OC (using fixed-order rotation) demonstrates superior generalization on training set. This indicates that the co-search of design thoughts via MCTS is more critical to final performance than fine-tuning the rotation order. We will strengthen this discussion in the revised manuscript.
>
> **7. Other multi-component co-design (KQ.5)**
> We thank the reviewer for highlighting these recent works. We clarify the fundamental differences below:
> - LaGO: The distinction lies in problem scope. LaGO is a SOTA optimizer for continuous black-box problems, hybridizing Bayesian optimization and trust-region methods to find solutions. However, E2OC belongs to the AHD and its output is a operator combination, rather than a solution for a specific problem instance.
> - G-LNS: Another LLM-based co-design paradigm. G-LNS focuses on destroy–repair pairs within a fixed LNS framework, using a synergy matrix and joint crossover for these coupled pairs. E2OC targets a broader suite of interdependent MOEA operators (e.g., crossover, mutation, and local search). It evolves operators at both the code level and the strategic level via MCTS, dynamically planning their roles and interplay to create a multi-level, adaptive algorithmic system.
>
> In the revised manuscript, we have expanded the "Related Work" to include a systematic comparison with LaGO and G-LNS. Due to their very recent release, direct experimental comparisons are not feasible for this revision. In the future, we will release the source code, prompts, and data for E2OC to facilitate community validation.
>
> All newly added tables can be found in ***Reviewer 3ag4 Supplementary*** or link.
> We hope the above explanation answers your question. Please feel free to ask if you have any further questions. Thank you.

---

> > ### Author Rebuttal · Reviewer_UaRm · 2026-04-03
> >
> > Thank you for the detailed rebuttal and for adding results. While the added clarifications are helpful, it remains unclear whether retained operators truly implement the selected design thoughts, since invalid or drifted generations seem to be filtered through executability and HV rather than directly checked for semantic adherence. So I maintain the score.

---

> > > ### Author Response · Authors · 2026-04-04
> > >
> > > Thank you very much for your affirmation of the added clarifications. We would like to provide further explanation regarding your question about semantic adherence.
> > >
> > > **E2OC selects design strategies based on performance rather than semantic adherence:**
> > > The objective of E2OC is to co-design efficient MOEA operator combinations under a limited budget, not to verify semantic adherence. We adopt a performance-oriented standard based on actual HV, aligning with prevailing AHD practices (FunSearch [1], EoH [2], MCTS-AHD [3], ReEvo[4]). In multi-operator systems, systemic performance depends on the combined thoughts of all operators; thus, **the semantic adherence of a single operator does not guarantee optimal global results**. In the Bi-FJSP experiment, we generated over 15,500 operators based on their combined actual optimization performance and code executability, rather than analyzing the degree of semantic adherence individually. By using MCTS to explore the strategy space via performance feedback, we **avoid the computational overhead of explicit semantic verification**.
> > >
> > > **Semantic adherence is not the focus of this work:**
> > > As addressed in our response to **Weakness 3**, semantic understanding remains a fundamental challenge across the AI and NLP fields. E2OC leverages systematic design strategies within prompts to guide the generation of high-performing operator combinations. Extensive research—including Chain-of-Thought (CoT) [5], multimodal models [6], and reflection-based AHD [4]—demonstrates that refining prompt information significantly enhances LLM capabilities. However, providing a definitive explanation for semantic adherence remains difficult due to the challenges of numerical mapping and the "black-box" nature of LLMs, even for G-LNS. This theoretical gap exceeds the scope of multi-operator co-design; nonetheless, **any future advancements in semantic adherence will directly reinforce the foundations of E2OC**.
> > >
> > > **Comparative experiments on semantic adherence:**
> > > In addition, we have conducted comparative experiments related to semantic adherence, which should effectively address your concerns. As shown in Table 2 and Appendix J.1, The LLM method uses a multi-agent framework to directly analyze and improve the design thought and the operators generated based on it. The agent improves the design thought of different operators based on decision information such as the **completion status of multiple operator suggestions** (semantic adherence), operator performance, and population features(e.g., code accuracy, performance distribution). However, the experimental results indicate that the performance of this method is significantly lower than that of E2OC across all test problems (see the search trajectory in Fig. 10 and the performance comparison in Table 2). This demonstrates that for the complex problem of multi-operator co-design, **relying on LLM for explicit semantic verification is less efficient than the MCTS search paradigm based on performance feedback**. This natural hierarchy of design strategies is better equipped to handle problem complexity.
> > >
> > > **Semantic consistency analysis of results:**
> > > Furthermore, although E2OC does not perform semantic verification during the process, its optimal output operators exhibit consistency with the design thoughts at the functional level. For example, in Bi-FJSP (Table 11 and Fig 16-17):
> > > 1. Thought $p_{1-1}$ (Fitness-dependent exploration): The operator uses *calculate_alpha()* to blend solutions based on fitness, adaptively balancing quality and diversity. While labeled "crossover," it effectively performs exploration similar to mutation. It also preserves the Pareto front by using *fitness1 - fitness2* to avoid blind averaging.
> > > 2. Thought $p_{3-1}$ (Performance-aware & diversity): The code uses *common_mask* and *divergent_mask* to identify identical and differing genes, performing crossover only in divergent regions. This precisely reflects "performance-aware" behavior—preserving high-quality segments while optimizing variable parts. It also uses *swap_mask* (50% probability) to maintain diversity and prevent stagnation.
> > >
> > > The explanations, experiments, and final code provided above regarding semantic adherence should be sufficient to resolve your concerns. We will also supplement the setup and analysis related to the semantic adherence aspects of the LLM method in the revised manuscript. Finally, we sincerely thank you for your thorough review of this work and hope to receive your further positive evaluation. Thank you again sincerely.
> > >
> > > [1] Funsearch. Nature, 2024.
> > >
> > > [2] EOH. ICML, 2024.
> > >
> > > [3] MCTS-AHD. ICML 2025.
> > >
> > > [4] Reevo. 2024.
> > >
> > > [5] Chain-of-thought prompting elicits reasoning in large language models. NIPS, 2022.
> > >
> > > [6] Multimodal LLM-assisted Evolutionary Search for Programmatic Control Policies. ICLR, 2026.

---

### Official Review · Reviewer_x4Ld · 2026-03-14

**Soundness:** 4
**Presentation:** 3
**Significance:** 3
**Originality:** 4
**Overall Recommendation:** 5
**Confidence:** 3

**Summary:**

This paper proposes E2OC, an LLM-based framework for the multi-operator co-design of interdependent operators in MOEAs for MCOPs. The key idea is to move beyond single-operator evolution and instead search for effective operator combinations by modeling the interdependencies and synergies among operators through combinations of design thoughts. E2OC performs the co-evolution of design strategies and executable codes, uses MCTS to progressively search the multi-domain design thought space, and applies operator rotation evolution to sequentially redesign and evaluate operators within an operator combination.

**Compliance With Llm Reviewing Policy:**

Affirmed.

**Final Justification:**

The author's response is sufficiently persuasive; therefore, the original evaluation remains unchanged.

**Key Questions For Authors:**

1. Since MCTS is used to explore the space of operator combinations, how well does the framework scale when the number of operators becomes much larger? In particular, could the search become intractable as the tree depth and branching factor increase?

2. The current experiments mainly consider TSP and FJSP. Could the authors provide evidence that E2OC generalizes to other multi-objective combinatorial optimization problems (MCOPs)?

3. The paper emphasizes the importance of interdependencies among operators, but I would like to see a more concrete analysis of the synergistic effects learned by E2OC. For example, what types of complementary operator interactions are actually discovered in the final operator combinations?

**Limitations:**

- The paper offers limited analysis of how sensitive E2OC is to the choice of backbone LLM and to the initialization of the multi-domain design thought space.

- Although the paper argues for the superiority of multi-operator co-design, the empirical study could be strengthened by comparisons against stronger baselines designed to challenge this claim more directly.

**Strengths And Weaknesses:**

(Strengths)

1. The co-evolution of design strategies and executable codes is a novel and meaningful idea.

2. The overall E2OC framework is systematic, and the roles of components such as MCTS and operator rotation evolution are clearly delineated.

3. The paper includes extensive empirical evaluation, including diverse baseline comparisons and rich ablation studies.

(Weaknesses)

1. The framework may be expensive in practice because it relies on repeated LLM API calls.

2. The experimental validation is limited to a relatively narrow range of MCOP instances.

3. The comparison may be somewhat favorable to E2OC, since it performs multi-operator co-design whereas several baselines only optimize operators independently under a narrower design scope.

---

> ### Author Rebuttal · Authors · 2026-03-31
>
> We appreciate the reviewer’s valuable feedback; our point-by-point responses are provided below:
>
> **1. Expensive in practice (Weak.1 & Lim.1):**
> Cost concern is a common challenge for LLM-driven AHD methods. E2OC’s costs arise from evaluation runtime and API fees, but remain low:
> - Quantification: For Bi-FJSP (Table 4), DeepSeek-Chat handles the co-design task for \\$1.2, generating 15,500 operators (Table S1). The per-operator cost of $7\times 10^{-5}$ is far below that of human experts.
> - Reduction Strategies:
> Offline computational overhead is halved by using a lightweight evaluator (as Table 7). Furthermore, employing local open-source models (* in Table 4) can reduce costs, with increasing practical value as these models evolve.
>
> **2. Generalization on more MCOPs (Weak.2 & KQ.2):**
> To better demonstrate E2OC's generalization ability on new problems and address concerns about overfitting, we have conducted the following additional analyses and experiments:
> - More Problems: Added Bi-KP and Bi‑CVRP tests. E2OC shows strong performance on these new problems (Table S3).
> - Larger scale: Operators designed on TSP100 maintain excellent performance on TSP150/200 (Table 10).
> - Overfitting: Training on instances of varied sizes and testing on different‑size held‑out instances (Table S4) confirms that learned operator synergies generalize well.
>
> These experimental results confirm that E2OC has strong generalization ability and robustness.
>
> **3. Baseline Fairness and Strength (Weak.3 & Lim.2):**
> - Fairness: To directly challenge the superiority of co-design, we integrated advanced single-heuristic AHD methods into different co-design frameworks. Table 2 shows that simply granting these methods co-design capability is insufficient, highlighting the necessity of E2OC’s strategy search and rotation evolution framework.
> - Strong Baselines: Compared to SOTA paradigms like Win-UCB (dynamic prompt rewriting), E2OC achieves similar training performance but significantly better test set generalization (Tab.2). We will open-source our code to facilitate community validation.
>
> **4. Scalability with more operators (KQ.1):**
> Tree depth of thought space grows linearly with the number of operators $K$, and the number of design strategy combinations grows as $AP^K$. And E2OC ensures tractability through a "focused search in a bounded space" strategy:
> - Complexity: Unlike methods that dynamically generate unbounded nodes (MCTS_Tuple), E2OC pre-constructs a set of $AP$ high-quality thought candidates for each operator by analyzing elite performers. This fixes the branching factor, preventing the tree from becoming intractable as $K$ increases.
>
> Tree complexity of MCTS variants: OC>Tuple>Sample=E2OC. Table 5 shows E2OC's superior performance, proving that searching a curated, high-quality subspace is scalable and efficient than exploring an unbounded, dynamic space.
>
> **5. Concrete Analysis of Operator Interdependencies (KQ.3):**
> While mathematical mapping remains an open challenge, we provide multi-level empirical evidence to reveal the specific synergies discovered by E2OC:
> - Coupling verification: Experiments with classic TSP operators (Tab.9) show that different sequences of the same operator set lead to significant performance variance. This confirms that E2OC exploits complex structural dependencies beyond individual operator effects.
> - Complementary thought: Ablating the thought of single operator in the final combination causes overall performance degradation (Tab.S5). This proves the set is a functionally organic whole at the strategic level, rather than a collection of independent ideas.
> - Code Analysis: Analysis of generated code (Fig 16-17) reveals a clear division of labor: E2OC-designed operators specialize in adaptive adjustments for machine genes while minimizing perturbation to the operation genes. This targeted interaction across chromosome components exhibits concrete, code-level synergy.
>
> E2OC discovers synergies through sequential and structural coupling, strategic interdependency, and component-specific functional specialization. Quantitative modeling of these interactions remains a key direction for future work.
>
> **6. Sensitivity to LLM and Initialization (Lim.1):**
> - LLM: Table 4 shows that cost-effective models and even locally deployed open-source models consistently outperform human-designed operators, proving E2OC's effectiveness is not tied to a specific LLM.
> - Poor Initialization: In a stress test using the worst-performing classic operators (Table 9) as the initial template, E2OC still evolved a combination that surpassed the best human-designed version (2-opt). Notably, this evolved set achieved a 30.93% and 22.06% lead on larger TSP150/200 instances (Table 10), confirming E2OC’s ability to escape poor initial states and achieve strong generalization.
>
> All newly added tables can be found in ***Reviewer 3ag4 Supplementary*** or link.
> If you have any further questions, please feel free to ask. Thank you.

---

> > ### Author Rebuttal · Reviewer_x4Ld · 2026-04-03
> >
> > The author's answer has resolved my questions.

---

> > > ### Author Response · Authors · 2026-04-08
> > >
> > > Thank you again for the careful reading and constructive discussion. We appreciate the feedback and will incorporate these clarifications in the final revision.

---

### Decision · Program_Chairs · 2026-04-30

**Decision:**

Accept (regular)

**Comment:**

The reviewers agreed that this paper addresses an important limitation of prior LLM-based automated heuristic design by moving from single-operator optimization to the co-design of interdependent operator sets in MOEAs. They also found the problem well motivated, the overall framework technically coherent, and the empirical results strong relative to expert-designed operators and existing AHD baselines. I have read the rebuttal and follow-up author comments carefully and taken them into account in this recommendation. The added analyses on significance testing, budget parity, broader generalization, initialization sensitivity, and operator interdependence substantially strengthened the paper and resolved most of the initial concerns.

A remaining concern is that the paper does not directly verify whether generated operators faithfully implement the selected design thoughts, since the search is ultimately driven by executability and downstream performance rather than explicit semantic adherence checks. This is a real limitation, and the final version should state this point carefully and avoid overstating the semantic interpretability of the method. However, the reviewers’ discussion also makes clear that this issue does not negate the paper’s main empirical contribution, namely that the proposed search procedure consistently finds stronger multi-operator combinations under a controlled evaluation budget.

Overall, the paper is technically solid, nontrivial in its design, and likely to be useful to a meaningful part of the ICML community working on LLM-based algorithm design and evolutionary optimization. While the semantic-level claims should be presented with more caution, the combination of novelty, practical relevance, and strengthened experimental support justifies weak acceptance.